



# Cryptotephra from the Icelandic Veiðivötn 1477 CE eruption in a Greenland ice core: confirming the dating of 1450s CE volcanic events and assessing the eruption's climatic impact

Peter M. Abbott[1], Gill Plunkett[2], Christophe Corona[3], Nathan J. Chellman[4], Joseph R. McConnell[4], John R. Pilcher[2], Markus Stoffel[5,6,7], Michael Sigl[1]

[1]Climate and Environmental Physics and Oeschger Centre for Climate Change Research, University of Bern, 3012 Bern, Switzerland
[2]Archaeology and Palaeoecology, School of Natural and Built Environment, Queen's University Belfast, Belfast BT7 1NN, UK
[3]Geolab, Université Clermont Auvergne, CNRS, F-63000, Clermont-Ferrand, France
[4]Desert Research Institute, Nevada System of Higher Education, Reno, Nevada 89512, USA
[5]Climatic Change Impacts and Risks in the Anthropocene (C-CIA), Institute for Environmental Sciences, University of
Geneva, 1205 Geneva, Switzerland
[6]Department of Earth Sciences, University of Geneva, 1205 Geneva, Switzerland
[7]Department F.-A. Forel for Environmental and Aquatic Sciences, University of Geneva, 1205 Geneva, Switzerland

*Correspondence to*: Peter M. Abbott (peter.abbott@climate.unibe.ch)

**Abstract.** Volcanic eruptions are a key source of climatic variability and reconstructing their past impact can improve our
understanding of the operation of the climate system and increase the accuracy of future climate projections. Two annually resolved and independently dated palaeoarchives – tree rings and polar ice cores – can be used in tandem to assess the timing, strength and climatic impact of volcanic eruptions over the past ~2,500 years. The quantification of post-volcanic climate responses, however, has at times been hampered by differences between simulated and observed temperature responses that raised questions regarding the robustness of the chronologies of both archives. While many chronological
mismatches have been resolved, the precise timing and climatic impact of one or more major sulphate emitting volcanic eruptions during the 1450s CE, including the largest atmospheric sulphate loading event in the last 700 years, has not been constrained. Here we explore this issue through a combination of tephrochronological evidence and high-resolution ice-core chemistry measurements from the TUNU2013 ice core.

We identify tephra from the historically dated 1477 CE eruption of Veiðivötn-Bárðarbunga, Iceland, in direct association with a notable sulphate peak in TUNU2013 attributed to this event, confirming that it can be used as a reliable and precise time-marker. Using seasonal cycles in several chemical elements and 1477 CE as a fixed chronological point shows that ages of 1453 CE and 1458/59 CE can be attributed, with a high accuracy, to two notable sulphate peaks. This confirms the accuracy of the NS1-2011 Greenland ice-core chronology over the mid- to late 15th century and corroborate the findings of
recent volcanic reconstructions from Greenland and Antarctica. Overall, this implies that large-scale Northern Hemisphere



climatic cooling affecting tree-ring growth in 1453 CE was caused by a Northern Hemisphere volcanic eruption in 1452 CE and then a Southern Hemisphere eruption, previously assumed to have triggered the cooling, occurred later in 1458 CE.

The direct attribution of the 1477 CE sulphate peak to the eruption of Veiðivötn, the most explosive from Iceland in the last 1,200 years, also provides the opportunity to assess its climatic impact. A tree-ring based reconstruction of Northern Hemisphere summer temperatures shows a cooling of −0.35 °C in the aftermath of the eruption, the 356[th] coldest summer since 500 CE, a relatively weak and spatially incoherent climatic response in comparison to the less explosive but longer-lasting Icelandic Eldgjá 939 CE and Laki 1783 CE eruptions, that ranked as the 205[th] and 9[th] coldest summers respectively. In addition, the Veiðivötn 1477 CE eruption occurred around the inception of the Little Ice Age and could be used as a 45 chronostratigraphic marker to constrain the phasing and spatial variability of climate changes over this transition if it can be traced into more regional palaeoclimatic archives.

## 1. Introduction

### 1.1 Assessing the timing, strength and climatic impact of volcanic eruptions

Volcanic eruptions are a key source of natural climatic variability, with the sulphate aerosols they emit into the stratosphere 50 shielding the Earth from solar radiation, causing short-term cooling from a local to global scale (Robock, 2000; Sigl et al., 2015). Reconstructing climatic variability caused by past volcanic forcing improves understanding of responses, feedbacks and internal variability in the climate system that can help increase the accuracy of future climate predictions (Schurer et al., 2013; Bethke et al., 2017). Two key proxies provide evidence for these reconstructions: tree rings, that record terrestrial climatic changes that could be related to volcanic eruptions, and polar ice cores, that record the atmospheric burden of 55 sulphate aerosols produced by past volcanic eruptions. Both archives have independent annually resolved chronologies that can be used to determine the timing of past volcanic events and their climatic impact. However, the quantification of post-volcanic climate responses have been hampered by differences in simulated and reconstructed temperature changes and potential mismatches in timing between the archives, which raised questions about the robustness of both tree-ring and ice-core chronologies (e.g. Baillie, 2008; Anchukaitis et al., 2012; Esper et al., 2013; Büntgen et al., 2014). While most of the 60 above-mentioned mismatches have been resolved (Sigl et al., 2015; Stoffel et al., 2015; Wilson et al., 2016), several enigmas remain. The most notable of these unsolved questions is the apparent discrepancy between the timing of one or more major sulphur-emitting volcanic eruptions, including the largest sulphate loading event in the last 700 years, and large-scale climatic cooling observed during the 1450s CE (Esper et al., 2017). This time period also aligns with the inception of the 'Little Ice Age' (LIA) in the Northern Hemisphere (Miller et al., 2012).


The large sulphate loading eruption has most commonly been attributed to the formation of the submarine Kuwae caldera offshore from Vanuatu in the South Pacific. This attribution is based on radiocarbon dating of charcoal that places the event,



first described in oral histories of local islands (see Ballard, 2020), in the mid-15th century (Monzier et al., 1994; Robin et al., 1994) and petrological evidence that it was a sulphur-rich eruption (Witter and Self, 2007). This attribution has become
commonly accepted within the literature despite a lack of confirmatory evidence. In recent studies, however, this source has been challenged or not attributed to the event (e.g. Németh et al., 2007; Esper et al., 2017; Toohey and Sigl, 2017; Hartman et al., 2019).

## 1.2 Timing of volcanic eruptions in the 1450s CE

Through the study of bristlecone pines in the western USA, LaMarche and Hirschboeck (1984) identified a series of frost
rings, including a widespread frost ring in 1453 CE, that they attributed to large volcanic eruptions. For the same year, Filion et al. (1986) and Delwaide et al. (1991) identified light rings in more than 25% of subfossil black spruce trees sampled in subarctic Quebec. Briffa et al. (1998) subsequently identified a distinct cooling event in 1453 CE in their network of tree-ring chronologies from the northern boreal forest while Salzer and Hughes (2007) identified evidence for anomalous tree-ring growth in the early and late 1450s CE in western USA bristlecone pines. More recently, warm season temperature
reconstructions from a Northern Hemisphere tree-ring network revealed spatially coherent and exceptional cooling in 1453 CE (Stoffel et al., 2015; Guillet et al., 2017), with cooling ranging spatially between −0.4 and −6.9°C, marking the onset of a 15-year cold period (Esper et al., 2017).

Various glaciochemical studies on Antarctic ice cores also provided evidence for a large volcanic event around 1450 CE
(e.g. Zanolini et al., 1985; Legrand and Kirchner, 1990; Moore et al., 1991). Delmas et al. (1992) suggested that this so-called "1450 event" was derived from a Southern Hemisphere eruption, based on a limited coeval signal in Greenland (Hammer et al., 1980), and associated it with the climate cooling in the tree-ring records to attribute it an age of 1452 CE. Langway et al. (1995) subsequently identified a bipolar volcanic event to which he attributed an age of 1459 CE and Zielinski (1995) attributed this to the large mid-15th century volcanic event. Later, in their compilation of 33 ice cores from
Greenland and Antarctica, Gao et al. (2006) suggested that the 1459 CE signal was from an eruption local to Greenland and concluded that the major Southern Hemisphere volcanic eruption occurred in late 1452 CE or early 1453 CE. However, this study overlooked that many of the ice-core chronologies from Antarctica were fixed to the tree-ring minima of 1453 CE following Briffa et al. (1998). Within the recent ice-core based independent volcanic reconstructions of Plummer et al. (2012) and Sigl et al. (2013, 2014, 2015) from both Greenland and Antarctica, two distinct sulphate peaks can be observed in
the 1450s CE (Fig. 1). The bipolar expression of these sulphate signals indicates that a comparatively smaller volcanic eruption, with a sulphate burden similar to that following Krakatau 1883 CE, occurred in the Northern Hemisphere low latitudes in 1452/53 CE and was followed by a massive low latitude Southern Hemisphere event in 1458/59 CE (Fig. 1; Sigl et al. 2013). Sulphur isotope analysis of both peaks in ice cores from Greenland and of the 1458/59 CE peak in Antarctica shows they both injected sulphur gases into the stratosphere and thus had the potential to impact global climate (Baroni et al.,
2008; Cole-Dai et al., 2013; Gautier et al., 2019).



Sigl et al. (2013) argued that the first eruption in 1452/53 CE caused the major climatic changes recorded in the tree-ring archives, that 1458/59 CE is the correct age for the large mid-1450s CE sulphate peak recorded at both poles and that this event contributed towards the decadal persistency of the cooling, but triggered less distinct climatic changes that were only

recorded in some tree-ring records (e.g. Salzer and Hughes, 2007). Subsequently, based on the new ages from the ice cores and observations from their tree-ring network, Esper et al. (2017) proposed two scenarios for the 1450s CE volcanic event: either the major Southern Hemisphere sulphate loading event did not cause large-scale cooling in the Northern Hemisphere extratropics after 1458 CE or the original ice-core age of 1452 CE proposed by Delmas et al. (1992) was correct and the cooling observed in the tree-ring records was caused by this event. Esper et al. (2017) suggest that the latter scenario is more

likely and contend that the dendrochronological evidence Sigl et al. (2013) used to support their dating to 1458/59 CE provides insufficient support for the new age. Here we further explore this debate and the scenarios proposed by Sigl et al. (2013) and Esper et al. (2017) by testing the accuracy of the recent ice-core chronologies using a tephra horizon from a historically dated Icelandic eruption.

## 1.3 Determining the source of volcanic eruptions in ice cores

While comparisons of the relative peak sulphate concentrations between ice cores from Greenland and Antarctica can be used to assess if volcanic events occurred in the Southern or Northern Hemisphere extratropics or in the tropics (Sigl et al., 2015), sulphate records alone cannot pinpoint the exact source of an eruption. Particles of volcanic ash (glass tephra shards) may be preserved in ice cores following eruptions in association with sulphate peaks and their chemical composition is a fingerprint of their volcanic source. Through geochemical analysis of individual shards and comparisons to characterisations

of proximal deposits it is often possible to determine the volcanic source of the ice-core tephra horizons (Abbott and Davies, 2012). The fine-grained nature, low concentration of particles and lack of visible expression (i.e. cryptotephras) of many of these deposits makes their identification and subsequent geochemical analysis challenging. However, a range of methods for the preparation and analysis of ice-core tephra samples has been developed and successfully applied to ice cores from Greenland and Antarctica (e.g. Dunbar et al., 2003; Davies et al., 2010; Dunbar and Kurbatov, 2011; Coulter et al., 2012;

Bourne et al., 2015, 2016; Iverson et al., 2017; Cook et al., 2018; Narcisi et al., 2019).

Identifying tephra in ice cores in direct association with sulphate peaks allows the volcanic source of those emissions to be determined with a high certainty; however, it has been shown that tephra shards are not found in direct association with all sulphate peaks (e.g. Abbott et al., 2012; Coulter et al., 2012; Bourne et al., 2015). Despite this, any tephra-based source

attributions are useful, especially for sulphate peaks assumed to link to known historical eruptions that provide keystone time markers for constructing, testing and synchronising ice-core chronologies (e.g. Öræfajökull 1362 CE – Palais et al., 1991; Laki 1783 CE – Fiacco et al., 1994; Katmai 1912 CE – Coulter et al., 2012). In addition, if such events occur in close association with other sulphate peaks of interest the dating control for all the events will be improved as the relative age





differences between closely-spaced peaks are more precise than absolute age estimates; so-called differential dating
(Andersen et al., 2006). This approach has been used to resolve the timing, strength and climatic impacts of past volcanic events; for example, the identification of tephra from the 'Millennium Eruption' of Changbaishan in Greenland (Sun et al., 2014) has helped to resolve these factors for that event and the earlier Icelandic Eldgjá eruption (Oppenheimer et al., 2017, 2018).

Here we explore whether tephra from the historically dated 1477 CE eruption of Veiðivötn- Bárðarbunga, Iceland, to which an obvious sulphate peak is commonly correlated and used as age-marker in Greenland ice-core chronologies (Fig. 1; Sigl et al., 2013, 2015), can be identified in a Greenland ice core. Direct attribution of this peak to the historical event will test and help improve ice-core dating around this marker and contribute towards the debate regarding the strength and climatic impact of volcanic eruptions during the 1450s CE. In addition, attribution of the sulphate peak to the 1477 CE Icelandic
eruption would provide the opportunity to explore the climatic impact of sulphate emitted during this eruption, which was a notable event as it is considered the most explosive Icelandic eruption during historical times, with a volcano explosivity index (VEI) of 6.

## 2. Veiðivötn 1477 CE

Iceland is the primary source of tephra horizons identified in the Greenland ice cores (e.g. Abbott and Davies, 2012; Bourne et al., 2015). As such, due to the high activity of many Icelandic volcanoes during the historical period, when eruptions can be precisely dated to the year, month, day and even hour using documentary records, they can provide key markers to test ice-core chronologies for the last ~1,000 years (Larsen et al., 2014). The Veiðivötn-Bárðarbunga volcanic system was one of the most active during the historical period, with notable eruptions recorded in 1797 CE, 1717 CE, 1500 CE, 1477 CE,
~1410 CE and ~1159 CE (Larsen et al., 1998, 2014). Of these events the eruption in 1477 CE was the most explosive and widespread, erupting over 10 km$^3$ of basaltic tephra from a 60–65 km long fissure along the Veiðivötn lake basin and distributing ash over about half of Iceland to the E, NE and N of the fissure (Larsen et al., 2014; Fig. 2b). Tephra from the 1477 CE eruption has been identified at many proximal and medial sites in Iceland as a visible horizon, including in soil sections, lake sediment cores and in offshore marine cores (Fig. 2b). Two distal cryptotephra occurrences have been
identified thus far (Fig. 2a), one in An Loch Mór, western Ireland (MOR-T1; Chambers et al., 2004) and another in Lake Getvaltjärnen, Sweden (Davies et al., 2007).

Early descriptions of deposits proximal to Veiðivötn did not ascribe a historical date to this eruption as it was thought to have not been mentioned in written sources; however, it was assigned an age of 1480 CE ± 11 years based on soil
accumulation rates between dated tephra layers (Larsen, 1984). Subsequently, a geochemical correlation was found between the Veiðivötn deposits and tephra layer "a", a widespread historically dated deposit previously thought to have originated from the Kverkfjöll volcanic system (Thórarinsson, 1958; Benjamínsson, 1981; Larsen et al., 2013). Thórarinsson (1958)





associated tephra layer "a" with an observed tephra-fall in the Eyjafjörður area, North Iceland (Fig. 1) that had been subsequently reported in a votive letter written on 11th March 1477 CE. Based on this a historical date of early 1477 CE,
most likely February, can be ascribed to the Veiðivötn-Bárðarbunga eruption and it is henceforth referred to as V1477 (G. Larsen, pers. comm., 2020).

## 3. Materials and Methods

### 3.1 TUNU2013 ice-core chemistry and chronology

The 213 m-long TUNU2013 ice core was retrieved from northeast Greenland during summer 2013 (78.04°N, 33.88°W; Fig. 2a) and covers the period between 280 CE and 2013 CE. Ice-core sections were measured at the Desert Research Institute (DRI) using a continuous melter analysis system that includes two Element2 (Thermo Scientific) high resolution inductively coupled plasma mass spectrometers operating in parallel to provide directly co-registered measurements of a broad range of ~35 elements, including sulphur, calcium and sodium (McConnell et al., 2002; Sigl et al., 2015; Maselli et al., 2017). There
are several sources of sulphur in ice core records, and therefore the sea salt component was removed to isolate the non-sea-salt (nssS) concentrations (Sigl et al., 2013). In addition to the elemental measurements, co-registered measurements of insoluble particle concentrations, in the size ranges of 2.6–4.5 and 4.5–9.5 µm diameter, were made using an inline Abakus® laser-based particle detector (Ruth et al., 2003). The nssS and fine particle concentrations were used to guide the tephra sampling towards depths with coeval peaks in both parameters. While the particle measurements are not a direct proxy or
solely controlled by the presence of tephra shards, this approach has been used successfully to identify tephra horizons in both Greenland and Antarctic ice cores (e.g. Jensen et al., 2014; Dunbar et al., 2017; McConnell et al., 2017, 2020; Plunkett et al., 2020).

The data from TUNU2013 were synchronised to the annually counted and seasonally resolved NS1-2011 chronology of Sigl
et al. (2015) using a series of volcanic markers as tie-points, including the sulphate peak thought to relate to V1477. The period covering 1450–1500 CE could be identified between ~75.79–82.20 m and clear peaks in sulphate thought to relate to 1453 CE, 1458/59 CE and 1477 CE were observed (Fig. 3a). Annual layers also were counted in TUNU2013 between 1445–1485 CE using concentration data for elements that show clear seasonal cycles, namely calcium and sodium whose variations are driven by continental dust and oceanic sea spray deposition, respectively, and the nssS/Na ratio (Sigl et al.,
2013; Fig. 5). This annual layer count permits the full assessment of the ages of the two 1450s CE sulphate peaks in TUNU2013 using differential dating if the V1477 tephra is identified.

### 3.2 Cryptotephra analysis

The 1477 CE sulphate peak in TUNU2013 was accompanied by a coeval peak in particle concentration, thus archived ice from the interval between 78.515–78.655 m was sampled directly for tephra analysis to determine if it related to the V1477

eruption (sample code QUB-1965; Fig. 3b). Direct ice samples (QUB-1964, -1963 and -1962) also were taken for tephra analysis across a later peak in particle concentrations, which according to the NS1-2011 is centred around late 1478 CE and early 1479 CE, and precedes a minor sulphate peak in mid to late 1479 CE (Fig. 3b).

The direct ice samples were transferred to Nalgene bottles, melted at room temperature and the meltwater centrifuged to
concentrate any particulate material present. The concentrated meltwater was evaporated onto a frosted microprobe slide and covered with Buehler EpoxyCure 2 resin. The presence of glass tephra shards was assessed at 100x-400x magnification using a polarising light microscope and a significant concentration of shards were identified in QUB-1965. To section the glass shards and create a flat surface for geochemical analysis the slide was ground using 12 µm alumina powder then polished using 6, 3 and 1 µm diamond paste. The surface of the resin and exposed shards was then carbon coated prior to the
oxide concentrations of 11 major and minor elements within individual glass shards being analysed using a JEOL FEG SEM 6500F at Queen's University Belfast. The operating conditions outlined in Coulter et al. (2010) were utilised with Cl and P additionally measured using EDS and WDS respectively. Glass from the rhyolitic Lipari obsidian and the basaltic Laki 1783 CE tephra were analysed as secondary standards during the analytical period and compared to recommended values from Kuehn et al. (2011). The raw sample analyses and secondary standard analyses are provided in the supplementary
information (Tables S1 and S2).

Geochemical analyses from several occurrences of the V1477 tephra were collated for data analysis (Fig. 2; Table S3). To facilitate graphical and statistical comparisons between all datasets the geochemical data were normalised to an anhydrous basis (i.e. 100% total oxides) using eight elements consistently analysed in all studies. As such, the minor elements MnO,
$P_2O_5$ and Cl were not included and their concentrations were deducted from the analytical totals prior to normalisation (see Table S3). Statistical comparisons were made between tephra occurrences using the statistical distance ($D^2$) test described in Perkins et al. (1995, 1998) and the similarity coefficient (SC) function of Borchardt et al. (1972) (Tables S4 and S5).

### 3.3 Tree-ring based reconstructions of cooling induced by Icelandic eruptions

### 3.3.1 Impact of volcanic eruptions on Northern Hemisphere summer temperatures

In addition to the ice-core and tephra analysis, we quantified the cooling induced by the V1477 eruption, using the NVOLC v2 dataset of Guillet et al. (2020), and compared it to other major Icelandic eruptions of the Common Era. The NVOLC v2 dataset includes 13 tree-ring width (TRW) and 12 maximum latewood density (MXD) chronologies spread across the Northern Hemisphere (NH; for further details see Stoffel et al. (2015) and Guillet et al. (2017)). All chronologies in NVOLC v2 have either been calibrated against regional instrumental climate data and transferred into temperature units or have been
interpreted by the original authors as a temperature proxy. Here, a principal component regression (PCR) approach was chosen to transfer TRW and MXD data into NH summer (or JJA for June-July-August) temperature units, expressed as





anomalies with respect to the 1961–1990 reference period. A reduced space signal of the proxy records was then extracted using principal component analysis (PCA) resulting in a set of principal component (PC) scores and PC loading patterns. The first *n* PCs with eigenvalues >1 were retained as predictors to develop a multiple linear regression model. A multiple
cross validation using random calibration sets (bootstrapping) was applied to the PCR to estimate the skill of the reconstruction and confidence intervals around the reconstructed anomalies. Because each chronology had different lengths, an iterative nesting method was used to develop the temperature reconstruction. This procedure entails the sub-setting of the original dataset into complete data matrices without missing values, so-called nests.

In NVOLC v2, 23 nests have been adjusted to the common period of all series. The earliest spans the period 500–551 CE and is composed of six chronologies. The most replicated nest (1230–1972 CE) included 25 chronologies. The nested PCR was computed schematically following a 3-step procedure. In each nest, firstly, the number of predictor variables was reduced using a principal component analysis; secondly, the PCs with eigenvalues >1 were retained as independent variables within Ordinary Least Square (OLS) multiple regression models while a mean NH JJA temperature series, obtained from the
Berkeley Earth Surface Temperature (BEST) dataset over the period 1805–1972 CE, was used as a dependent variable. The robustness of each model was tested based on a traditional split calibration/verification procedure bootstrapped 1,000 times and the final reconstruction of each nest was computed as the median of the 1,000 realizations, given with their 95% confidence interval. The skill of each reconstruction has been evaluated based on the coefficient of determination ($r^2$ for the calibration and $R^2$ for the verification periods), reduction of error (RE) and coefficient of efficiency (CoE) statistics. The
final reconstruction (500–2000 CE) was achieved by splicing all the nested time series after the mean and variance of each nested reconstruction segment had been adjusted to the best replicated nest (1230–1972 CE). Finally, reconstructed temperature anomalies are presented as deviations from the 30-yr mean climatology around the year of the volcanic eruption. For example, for V1477, a background was calculated by averaging the windows 1462–1476 CE and 1478–1493 CE then an anomaly was created by subtracting this background from the 1477 CE reconstructed temperature.

**3.3.2 Regional variability of Northern Hemisphere summer cooling**

In order to characterize regional-scale climate changes and to reveal spatial anomaly patterns in the aftermath of Icelandic eruptions, we additionally used a network of tree-ring proxy data to develop a climate field reconstruction of extratropical NH JJA temperatures spanning the last 1,500 years. In this approach, we grouped the 25 chronologies from NVOLC v2 in 11 regional clusters using a minimum coefficient correlation (r) threshold above 0.3 (p<0.05) over the common period of the
chronologies. Based on the later threshold, we included only one chronology in the Quebec, Central Europe, Siberia–Taymir, Siberia–Yakutia, and China–Qilian Mountains clusters whereas five composed the Western Europe region. In the six clusters that included multiple chronologies, we derived reconstructions from bootstrapped nested PCRs. In the remaining five clusters, we used a bootstrap OLS linear regression to quantify interannual variations of summer temperatures. Based on



CoE statistics >0.1, 3,486 grid points have been reconstructed back to 500 CE. The smallest (Indigirka) and largest (Siberia–
Poral Ural) clusters included 53 and 918 grid points, respectively.

## 4. Results

### 4.1 Cryptotephra analysis

#### 4.1.1 1477 CE sulphate peak

A closer look at monthly (~1 cm) resolution data for the sulphate peak attributed to 1477 CE shows particle concentrations
peak in late winter 1477 CE before sulphate concentrations peak 2 months later during the spring and elevated
concentrations are broadly restricted to 1477 CE (Fig. 3b). Volcanic particles, including glass tephra shards, are always
transported and deposited rapidly (within days to weeks) following an eruption due to their relatively high density. However,
the longer residence time of sulphate aerosols means that these concentrations stay elevated for a more extended period
following a volcanic eruption, potentially for several years after a large eruption injecting aerosols high into the stratosphere
(Robock, 2000). The sharp and narrow peak in sulphate concentrations observed in 1477 CE (in contrast to 1458/59 CE; Fig.
1) and the short delay following the particle concentration peak is more typical of a Northern Hemisphere eruption local to
Greenland and the timing of their deposition early in 1477 CE is consistent with the historical date for the V1477 eruption.

Within the TUNU2013 ice sample covering 78.515-78.655 m depth (QUB-1965), ~126 tephra glass shards with a greenish-
brown blocky morphology and long axis diameters up to 30 µm were observed (Fig. 3c). The major element composition of
16 of these shards was analysed and they form a single homogenous population with a tholeiitic basaltic composition (Fig.
4ai). The most likely high-latitude Northern Hemisphere volcanic source of material of this composition is Iceland and
comparisons to the geochemistry of material proximal to several Icelandic volcanic systems shows it was most likely
285    sourced from the Veiðivötn-Bárðarbunga system (Fig. 4aii and iii).

Comparisons were made to the collation of geochemical analyses for proximal and distal occurrences of tephra produced
during the V1477 event. Graphical comparisons show that for most element oxides, e.g. $Al_2O_3$, $TiO_2$, $K_2O$ and $SiO_2$, there
are close similarities between the concentrations observed in the TUNU2013 shards and the proximal and distal V1477
290    occurrences (Fig. 4bi and ii). However, for some elements such as $Na_2O$, FeO and MgO, some offsets, most notably of up to
1 wt% for MgO, can be observed between the main populations (Fig. 4biii). Some individual analyses from the V1477
occurrences, which could be dismissed as outliers, do however have MgO concentrations similar to those from the
TUNU2013 78.655 cm shards (Fig. 4biii). The geochemical differences are reflected in statistical comparisons between the
TUNU2013 tephra and the collation of V1477 data (Tables S4 and S5). Statistical distance tests show that in comparison to
~36 % of the V1477 occurrences the composition of TUNU2013 78.655 m is statistically different at the 99 % confidence





interval and the D$^2$ values for several other comparisons are just below the critical value (Table S4). A matrix of similarity coefficient comparisons between the average compositions of the tephra characterisations shows that all of the comparisons of prior V1477 occurrences have high values above typical thresholds for accepting that characterisations of Icelandic basaltic eruptions are identical, with all comparisons >0.95 and the vast majority >0.97 (Table S5). In contrast, comparisons

between TUNU2013 78.655 m and the V1477 occurrences range between 0.917 and 0.957, values typically thought to indicate that the tephras originate from the same volcanic source but not the same eruption (Table S5; Begét et al., 1992; Abbott et al., 2018).

Geochemical offsets can result from tephra shards with the same composition being analysed under contrasting operating

conditions on different microprobes and/or at different times (Kuehn et al., 2011). However, an inspection of secondary standard analyses made during the analytical periods when the TUNU2013 shards and those from the Skaftártunga occurrence of V1477, reported in Streeter and Dugmore (2014), were analysed strongly indicates that the geochemical offsets are not the result of analytical issues but are real differences in composition between the sample datasets (Fig. S1).

Overall, despite the geochemical offset, the evidence supporting the correlation of the TUNU2013 78.655 m tephra layer to the V1477 event is strong. Firstly, the ice-core chemistry and the shard geochemistry indicate that TUNU2013 78.655 m was sourced from a Northern Hemisphere volcanic region close to Greenland and has a composition matching products of the Veiðivötn-Bárðarbunga volcanic system. Secondly, the geochemical differences between TUNU2013 78.655 m and the V1477 occurrences are only slight and for most elements the concentrations are highly similar. Possible explanations for the

differences in composition will be discussed in Sect. 5.4. Thirdly, while it is acknowledged that there can be uncertainties in ice-core chronologies, chronological uncertainties in this age range would be of the order of years rather than the tens to hundreds of years required for TUNU2013 78.655 m to be the product of one of the other known Veiðivötn-Bárðarbunga eruptions (e.g. V1717, V1500, ~V1410). In addition, of these eruptions V1477 had the largest magnitude and thus the greatest potential for widespread deposition of its products. Lastly, as the Icelandic tephrostratigraphy is well documented

over the last 1,000 years (e.g. Thordarson and Larsen, 2007; Larsen and Eiríksson, 2008), it is highly unlikely that TUNU2013 78.655 m was deposited following a previously unknown eruption of the Veiðivötn-Bárðarbunga volcanic system.

**4.1.2 1479 CE sulphate peak**

A limited number of tephra shards were found in the ice samples from the later microparticle peak at the start of 1479 CE,

with ~3 brown shards identified in both QUB-1963 and QUB-1962. A tephra layer with a rhyolitic composition was identified previously around 1479 CE by Fiacco et al. (1993) in the GISP2 ice core and attributed to an eruption from Mount St Helens, USA, dated using dendrochronology (Yamaguchi, 1983), although the geochemical match with this source is ambiguous (Abbott and Davies, 2012). The relationship between microparticle and sulphate concentrations for the 1479 CE





event in TUNU2013 matches the observations of Fiacco et al. (1993), which also showed a peak in microparticles preceding

a sulphate peak by about half a year (Fig. 3b). The tephra shards from QUB-1963 and QUB-1962 have not been geochemically analysed to determine if they correlate to the tephra layer reported in Fiacco et al. (1993). However, a geochemical correlation is highly unlikely as the physical characteristics of the TUNU2013 shards are indicative of a basaltic composition, not the rhyolitic composition of the 1479 CE tephra of Fiacco et al. (1993). Therefore, it is most likely that these shards represent remobilised shards possibly related to the basaltic V1477 layer; however, this does not rule out that

the 1479 CE sulphate peak in TUNU2013 derived from another eruption.

### 4.2 Differential dating of 1450s CE volcanic events

Through the identification of the V1477 tephra layer in TUNU2013 in direct association with the large sulphate peak at ~78.56 m, it is demonstrated that the prior assumption that it represents aerosol deposition from the 1477 CE eruption of Veiðivötn is correct. As such, the assertion that 1477 CE can be used as the age for this ice-core depth with 0 years

uncertainty is appropriate. Using this age as a fix-point for the annual layer counts for TUNU2013 between 1445 CE and 1485 CE it is shown that the ages of 1453 CE and 1458/59 CE for the two 1450s CE sulphate peaks are in agreement with those derived from the NS1-2011 chronology (Fig. 5). These ages were based on annual layers counts from NEEM-2011-S1 using an age of 1477 CE for a notable sulphate peak in that sequence (Sigl et al., 2015). The close correspondence between the nssS records from TUNU2013 and NEEM-2011-S1 and the strong annual signals in TUNU2013 provides further

evidence to support those ages for the events (Fig. S2).

### 4.3 Tree-ring based reconstruction of summer temperature cooling induced by Icelandic eruptions

Within the NVOLC v2 reconstruction, each nest passed all verification tests, $R^2$ and RE, for the calibration period, as well as $r^2$ and CoE for the verification period. For the 1420-1650 CE period (Fig. 6a), the unfiltered reconstruction indicates a cooling in the aftermath of V1477 of −0.35 °C relative to the 1961–1990 reference period, the 356[th] coldest NH summer

since 500 CE (i.e. rank 356). By comparison, stronger cooling is observed following the Icelandic Eldgjá 939 CE (−0.5°C in 940 CE, rank 205) and Laki 1783 CE (−1.2° C, rank 9) eruptions. More extreme negative anomalies are observed following other major explosive volcanic eruptions, including −1.6 °C in 1601 CE (rank 1, Huaynaputina), −1.5°C in 1816 CE (rank 3, Tambora), −1.4 °C in 536 CE (rank 4, unidentified volcano) and –1.3°C in 1453 CE (rank 5, unidentified volcano) (Fig. 6a). After filtering with a 30-yr running mean, the NH summer cooling prevailing in 1477 CE was −0.1°C (Fig. 6b), the 524[th]

coldest NH summer since 500 CE. Our findings are corroborated through comparisons with other recent, large-scale reconstructions, Sch2015 (Schneider et al., 2015) and NTREND2015 (Wilson et al., 2016), which rank 1477 CE as the 538[th] (−0.35°C) and 182[th] (−0.23°C) coolest summers, respectively (Fig. 6b). The Sch2015, NTREND2015 and NVOLC v2 reconstructions all show more significant cooling in the aftermath of the Eldgjá and Laki eruptions, with cooling ranging between −0.4 to −0.6 °C and −0.7 to −1°C, respectively (Fig. 6b). Grouping the 25 chronologies into 11 clusters enables an

estimation of the regional variability of summer cooling induced by the Icelandic eruptions (Fig. 6c). Based on this





approach, the JJA gridded temperature reconstructions for 940 CE and 1783 CE (Fig. 6c) indicate that volcanic forcing induced widespread cooling over the NH. Extremely cold conditions prevailed in Scandinavia, Western Europe and in Coastal Alaska in 940 CE, and on the Yamal Peninsula and in Alaska in 1783 CE. Less consistent patterns are observed in the reconstructed temperatures following the V1477 eruption, with cold temperatures in Scandinavia, Western Europe and

Central Asia, contrasting with above-average anomalies over North America, the Polar Ural and on the Yamal Peninsula (Fig. 6c).

## 5. Discussion

### 5.1 Confirming the timing of mid-15th century volcanic eruptions

The identification of the V1477 tephra layer in TUNU2013 has demonstrated the veracity of the age of 1477 CE commonly used for a keystone sulphate peak identified in many ice cores from Greenland. This provides evidence to support the accuracy of the NS1-2011 Greenland ice-core chronology around this period and corroborates the ages proposed by Sigl et al. (2013) for the mid-1450s CE sulphate peaks in the Greenland ice-core records deposited following large volcanic eruptions in 1452 CE and 1458 CE. The ages, temporal separation and duration of sulphate deposition for these events are

consistent with those for two volcanic eruptions recorded in the Antarctic WDC core, whose ages were derived using an independent annual-counted timescale and not bipolar synchronisation of the sulphate records (Sigl et al., 2013; Fig. 1). As highlighted by Sigl et al. (2013), the sulphate deposition in Greenland relating to the later event is greater than for the earlier event, and is also consistent with the record from Antarctica. Further independent confirmation of the matching of the events could be achieved if the temporal evolution of sulphur isotope anomalies over the 1458/59 CE period is consistent between

ice cores from both polar ice sheets, as was demonstrated recently for the Samalas 1257 CE eruption (Burke et al., 2019).

These findings demonstrate that the scenario proposed, but not favoured, by Esper et al. (2017), is correct, i.e. a strongly asymmetric Southern Hemisphere sulphate loading event occurred in 1458 CE without causing large-scale Northern Hemisphere cooling. In addition, the findings back up the argument of Sigl et al. (2013) that the first eruption in 1452 CE

occurred in the Northern Hemisphere and caused the distinct climatic cooling observed in the tree-ring records. The close succession of the two large sulphur-rich volcanic eruptions was the most likely cause of the longevity of the 15-year cold period following 1452 CE.

### 5.2 Testing ice-core chronologies and determining volcanic source using tephra horizons

This study has shown the power of using historically dated tephra horizons to test the accuracy of ice-core chronologies and

assumptions regarding the source of sulphate aerosols deposited over ice sheets, which has critical implications for assessing the climatic impact of eruptions. The example provided here can be added to others from Greenland, with tephra from Öræfajökull 1362 CE, Laki 1783 CE and Katmai 1912 CE having all been identified in ice cores from Greenland in



association with chemical indicators previously attributed to those eruptions (Palais et al., 1991; Fiacco et al., 1994; Coulter et al., 2012). Determining the source of notable peaks in sulphate or other volcanic indicators (e.g. acidity and electrical

conductivity) using tephra is necessary, when possible, as associations to specific events can become embedded in the literature with limited supporting evidence, even if only originally proposed as a possibility. For example, associations between chemical indicators and the Hekla 1104 CE and Vesuvius 79 CE eruptions, that previously have been used as historical tie points with no uncertainty while building chronologies (Hammer et al., 1980; Zielinski et al., 1994; Vinther et al., 2006), have been found to be inaccurate following re-evaluation of the Greenland ice-core chronologies (Sigl et al.,

2015). The volcanic signal previously associated with Hekla 1104 CE has been re-dated to 1108 CE and Guillet et al. (2020) have recently proposed it can be linked to an eruption of Mount Asama, Japan; however, at present this is not supported by tephra evidence. The volcanic signal previously associated with Vesuvius 79 CE has been redated to 87/88 CE by Sigl et al. (2015). The re-evaluation of the Greenland ice-core chronology was aided by documented historical reports (Sigl et al., 2015) and cosmogenic radionuclides in tree rings and ice cores as another form of isochronous marker horizon, most likely

produced by solar particle events (Miyake et al., 2012, 2015; Mekhaldi et al., 2015; Büntgen et al., 2018). This highlights that tephra is just one of the tools that can be used to evaluate and improve ice-core chronologies.

Testing associations between volcanic indicators in ice cores and specific events using tephra could be useful for determining the source of the two 1450s CE eruptions, whose sources are currently unknown. The 1458 CE event often has

been attributed to the formation of the submarine Kuwae caldera, however, the evidence is inconclusive and circumstantial so the source is still open for debate. Hartman et al. (2019) characterised tephra from an Antarctic ice core found in association with the 1458/59 CE event and concluded that it was distinct to eruptive products from Kuwae; however, despite some geochemical similarities to the products of a local Antarctic volcano, a definitive source was not proposed. Based on the size of the particles and the expression of the sulphate in different ice cores, Hartman et al. (2019) suggest that it resulted

from a tropospheric aerosol cloud; however, this is inconsistent with sulphur isotope evidence from Antarctica for stratospheric injection of the sulphate cloud (Baroni et al., 2008; Gautier et al., 2019). A local tropospheric source is also inconsistent with the widespread deposition of volcanic sulphate that prevailed for 2–3 years as observed in a large network of 16 ice cores all over Antarctica (Sigl et al., 2014; Toohey and Sigl, 2017). Further studies of sulphur isotopes coupled with attempts to identify tephra associated with both 1450s CE events and more work on proximal records, could aid in

conclusively identifying their sources.

## 5.3 Use of V1477 as a chronostratigraphic marker for the inception of the LIA

The mid-15[th] century is widely considered as the inception of the LIA, that potentially was intensified by the large 1450s CE volcanic events (Miller et al., 2012; Schurer et al., 2014; Slawinska and Robock, 2018). As such, if the V1477 tephra layer could be traced into more palaeoclimatic archives it could act as a useful chronostratigraphic marker for constraining the

chronology of different proxy records and assessing the phasing and spatial variability of climatic changes associated with



the LIA onset. The identification of the V1477 tephra layer in a Greenland ice core corroborates the findings of past studies of terrestrial sequences from Ireland and Sweden (Chambers et al., 2004; Davies et al., 2007) that following the eruption its products were dispersed over a wide geographical area. Therefore, it has significant potential to act as a chronostratigraphic tie-line linking records from Greenland, the North Atlantic and Northwest Europe. As the most explosive Icelandic eruption

in the last 1,000 years, it is notable that V1477 has not already been identified at more sites and established as a key horizon in the tephrochronological framework for northern Europe (Lawson et al., 2012). The generally lower explosivity of Icelandic basaltic eruptions reduces the likelihood of the injection of material high into the atmosphere (Lawson et al., 2012), and the potential transport distance of the tephra is then dependent upon tropospheric conditions at the time of the eruption. Differences between glass shards produced during basaltic and rhyolitic eruptions, may further constrain the long-range

transport of basaltic glass tephra shards with the greater density, lower vesicularity, and generally larger grain-size making the shards more prone to rapid fallout (Watson et al., 2017). However, it seems unlikely that these factors can explain the lack of more widespread V1477 occurrences in NW Europe given the prior distal identifications and the known high explosivity of the eruption. It has been proposed that basaltic tephra shards are not identified in some terrestrial sequences as they can degrade in acidic peatland environments (Pollard et al., 2003) or are overlooked in lake sediments due to the

difficulties in separating and identifying basaltic shards (Lawson et al., 2012). In addition, non-identification could be due to (1) spatial heterogeneity in distal ash deposition (Davies et al., 2010), (2) latitudinal differences in wind strength, since although rarely identified in continental European sequences basaltic shards have been identified in a Russian Arctic site located a similar distance from Iceland but further north (Vakhrameeva et al., 2020), and/or (3) this specific time interval not being investigated. Therefore, we propose that the potential use of V1477 as a chronostratigraphic marker could be increased

through focused cryptotephra investigations of specific time windows and, for lake sediments, incorporating magnetic separation processing (e.g. Mackie et al., 2002) to maximise the isolation of basaltic material.

A key consideration for any future studies aiming to trace V1477 is the strong similarity between the geochemical compositions of V1477 and other notable Veiðivötn sourced tephra layers such as ~V1410, V1717 and V877, the Landnám

tephra (Cage et al., 2011; Harning et al., 2018). To date, only the V1477 and Landnám tephras have been identified definitively in European records, although the potential presence of tephra shards from other Veiðivötn eruptions (e.g. ~V1159, V1717, V1766 and V1797) has been discussed (Plunkett and Pilcher, 2018; Vakhrameeva et al., 2020). Therefore, in palaeoarchives with a lower temporal resolution than the ice cores independent chronological evidence would be required to support any correlations to specific Veiðivötn eruptions such as, the radiocarbon age Davies et al. (2007) used to support

the correlation of the Lake Getvaltjärnen deposit to V1477.

**5.4 Geochemical variability of V1477 deposits**



A notable observation from this work was a slight geochemical offset between tephra shards from TUNU2013 78.655 m and the V1477 occurrences (Fig. 4b). However, as outlined in Sect. 4.1.1 it does not hinder the attribution of TUNU2013 78.655 m to V1477. The offsets are most notable for the MgO concentrations which, except for a few outlier analyses from other deposits, are ~0.75-1 wt% higher in the TUNU2013 shards (Fig. 4biii). MgO concentrations around 7-8 wt% are not atypical for Veiðivötn products (Fig. 4aii), but are more characteristic of Veiðivötn eruptions that occurred earlier in the Holocene than 1477 CE (Óladóttir et al., 2011; Caracciolo et al., 2020).

The difference in MgO concentrations indicates that the TUNU2013 shards have a more primitive, i.e. less evolved, geochemical composition than shards from the other V1477 occurrences. A general trend of increasing MgO values with decreasing FeO concentrations can be observed in the V1477 data (Fig. 4biii) and is most obvious in the characterisation of V1477 from Skaftártunga (Streeter and Dugmore, 2014). Within the Skaftártunga characterisation, alongside shards with strong similarities to the other V1477 analyses, more geochemically evolved shards with lower MgO concentrations and higher FeO values than most of the other V1477 analyses can be observed as well as two shard analyses with MgO and FeO values comparable to the TUNU2013 78.655 m data (Fig. 4biii). This overall trend could be indicative of geochemical evolution during the eruption. The bilobate form of proximal V1477 deposits on isopach maps and differences in the size and morphology of tephra shards between the lobes indicates that differential dispersal occurred with plumes to the ENE and NNW of the fissure (Benjamínsson, 1981). It has been suggested that the two lobes resulted from a shift in wind direction from SW to SE during the eruption (Benjamínsson, 1981) or bifurcation of the volcanic plume due to a crosswind (Ernst et al., 1994). These processes could have resulted in different phases of the eruption being transported along the two dispersal axes and account for the slight differences between geochemical characterisations from specific sites.

As prior studies have not identified such geochemical differences, it should be acknowledged that they could have resulted from an analytical uncertainty not captured with secondary standard analysis when the TUNU2013 shards were analysed (Fig. S1). Further investigation of V1477 occurrences at proximal sites and in other Greenland ice cores could determine if the geochemical offset is consistent for other deposits along the same dispersal axis and the potential for offsets should be considered within any attempts to trace this horizon.

### 5.5 Climatic impact of V1477 relative to other Icelandic eruptions

The definitive attribution of the 1477 CE sulphate peak in Greenland to the Icelandic V1477 eruption also provided an opportunity to assess the climatic impact of this explosive eruption in comparison to the effects of other historical eruptions. Overall, despite having a VEI of 6, the NH cooling following V1477 was relatively weak and spatially incoherent in comparison to the less explosive Icelandic Laki and Eldgjá eruptions (VEI 4-5), and more explosive lower-latitude eruptions, e.g. Tambora and Huaynaputina (Fig. 6; Sect. 4.3). The greater impact of the more effusive Icelandic eruptions relative to the V1477 eruption can be attributed to the huge sulphate load emitted during the Laki and Eldgjá eruptions over prolonged



periods of time, ~8 months and >2 years, respectively, in comparison to the short-lived, days to weeks, duration of the V1477 eruption (Thordarson et al., 2001; Thordarson and Larsen, 2007). The short residence time of sulphate in the atmosphere following the V1477 eruption is highlighted by the sharp and narrow nature of the 1477 CE sulphate peak in the Greenland ice cores (Fig. 1). In addition, it is likely that sulphate from the V1477 eruption was not injected high into the

stratosphere as the latitudinal range of Iceland is affected by descending air masses that are strongest during the winter (Butchart, 2014), the season during which V1477 occurred.

## 6. Conclusions

Using sulphate and particle concentrations records as a guide, volcanic ash from the Icelandic eruption of Veiðivötn in 1477 CE has been identified as a cryptotephra in the TUNU2013 Greenland ice core. This identification verifies the widely assumed source of a notable sulphate peak, present in many Greenland ice cores, and confirms the accuracy of the NS1-2011 chronology around this depth. In addition, using annual layer counting the ages of 1452 CE and 1458 CE attributed to two volcanic events in the 1450s CE are confirmed with a high accuracy. While being considered the most explosive Icelandic

eruption of the last 1,200 years, a tree-ring based reconstruction of summer temperatures demonstrates that V1477 caused weak and spatially incoherent summer cooling in comparison to other recent Icelandic eruptions. The timing of the eruption around the inception of the LIA in the NH means V1477 could act as a chronostratigraphic marker for assessing climatic changes associated with this onset. However, potential geochemical variability between V1477 deposits and geochemical similarities with the products of other eruptions from Veiðivötn must be considered when correlations are being assessed.


### Data availability

The data presented and utilised in this study are available in the supplementary material.

### Author contributions

All authors contributed towards obtaining and analysing the data utilised in the study. JRM, NJC and MSi contributed ice-
core glaciochemical records. GP and JRP performed tephra geochemical analyses. MSt and CC produced the North Hemisphere tree-ring based summer temperature reconstruction. PA wrote the manuscript with MSt and CC contributing Sections 3.3 and 4.3 and Figure 6. All authors contributed towards improving and editing the manuscript.

### Competing interests

The authors declare that they have no conflict of interest.




## Acknowledgements

Thanks to Carla Grimaldi for compiling some of the V1477 geochemical data. Thanks to Guðrún Larsen for a detailed description of the history of the dating of V1477 and Esther Ruth Guðmundsdóttir for providing geochemical data from the V1477 occurrence in Lake Lögurinn. Thanks to Richard Streeter and Chris Hayward for providing secondary standard data
related to the V1477 occurrence from Skaftártunga.

## Financial support

PA and MSi received funding from the European Research Council under the European Union's Horizon 2020 research and innovation programme (grant agreement no. 820047). The collection and analysis of the TUNU2013 core was supported by the U.S. National Science Foundation (grant #1204176) to JRM, with additional support (grant #1925417) to JRM and NJC
for the volcanic interpretation. MSt and CC received funding from the Swiss National Sciences Foundation through the SNSF Sinergia CALDERA project (grant agreement no. CRSII5_183571).

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





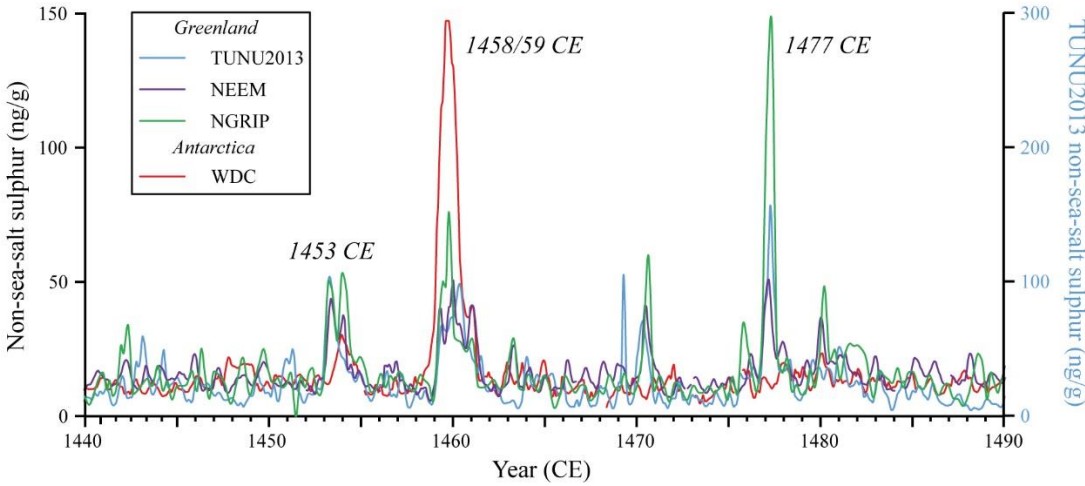

**Figure 1: Bipolar comparison of monthly resolution ice-core non-sea-salt sulphur concentrations between 1440–1490 CE. Greenland data from the NEEM-2011-S1 ice core (Sigl et al., 2013), the TUNU2013 ice core (Sigl et al., 2015) and the North Greenland Ice Core Project (NGRIP) ice core (Plummer et al., 2012) all plotted using the annually counted NS-2011 chronology (Sigl et al., 2015). Antarctic data is from the annually counted West Antarctic Ice Sheet Divide ice core (WDC) on the WD2014 age scale (Sigl et al., 2016).**

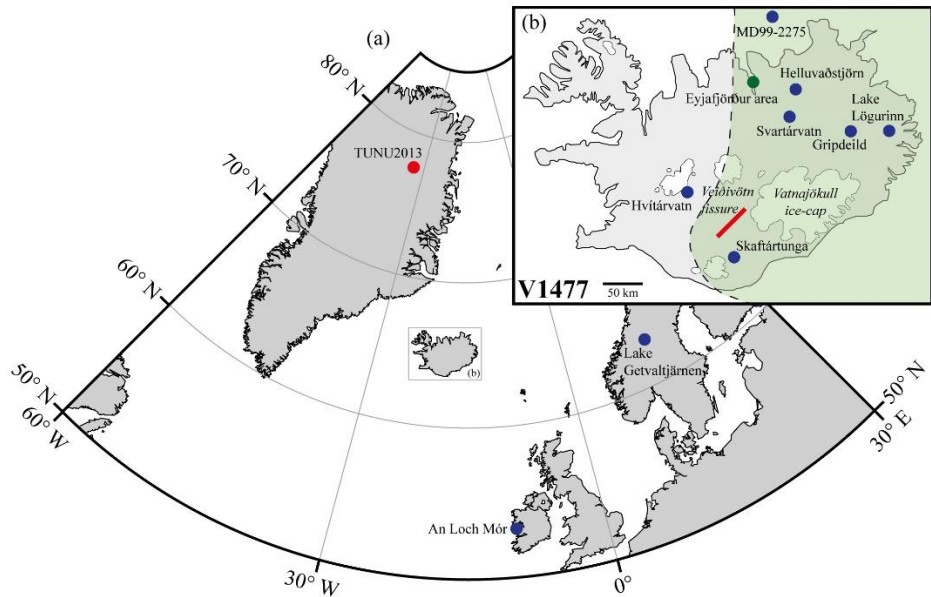

**Figure 2: (a) Location map of the TUNU2013 ice-core drilling site and sites outside Iceland containing tephra attributed to the Veiðivötn 1477 CE eruption. (b) Location map of Icelandic sites containing tephra from the Veiðivötn 1477 CE eruption, from which geochemical characterisations were used in this study, and the 60–65 km long fissure active during the 1477 CE eruption. The green shaded area covers the 0.5 cm isopach of the Veiðivötn 1477 CE eruption. Adapted from Larsen et al. (2014).**


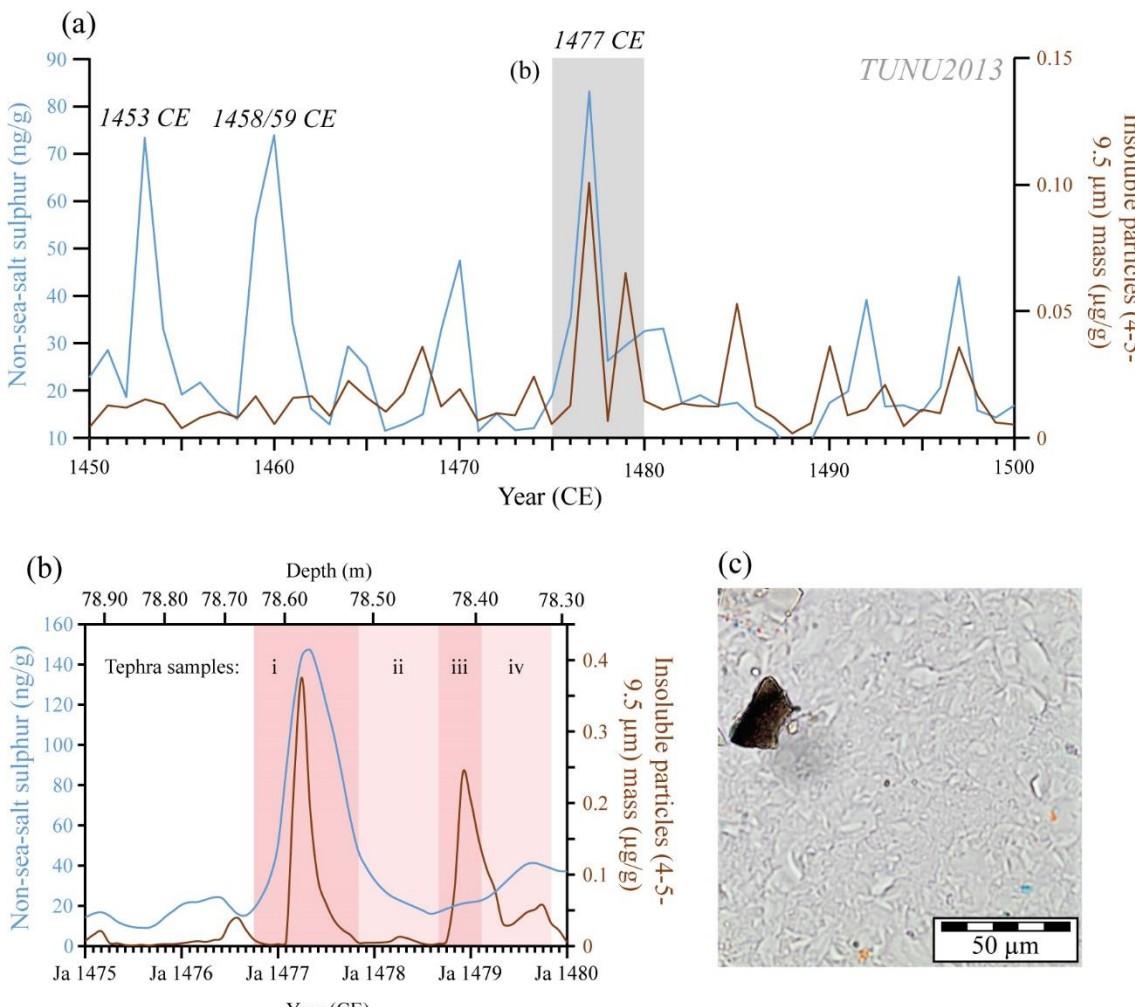

**Figure 3: (a) Annual resolution sulphur and particle concentration data from TUNU2013 between 1450–1500 CE. Data plotted using the NS1-2011 chronology. (b) Monthly resolution sulphur and particle concentration data from TUNU2013 between 1475–1480 CE. Data plotted using the NS1-2011 chronology. Red bars denote depth intervals of tephra sampling. Tephra samples: (i) QUB-1965 = 78.515–78.655 m (ii) QUB-1964 = 78.435–78.515 m (iii) QUB-1963 = 78.395–78.435 m (iv) QUB-1962 = 78.315–78.78.395 m. (c) Example of a glass tephra shard identified in the TUNU2013 78.655 m deposit.**



(a) (i)

Trachybasalt

Alkaline series

Tholeiitic series

Basaltic    Basaltic andesite

(b) (i)

(ii)

(iii)

● TUNU2013 78.655 m (QUB-1965)

**Icelandic volcanic systems:**

Veiðivötn-Bárðarbunga    Grímsvötn

Kverkfjöll    Askja

**Veiðivötn 1477 CE occurrences:**

◆ V1477, Svartárvatn    ✖ V1477, Gripdeild

✚ V1477, Helluvaðstjörn    ⬡ V1477, Vatnajökull comp.

◆ V1477, Skaftártunga    ✚ MD99-2275 179 cm

⬟ V1477, Hvítárvatn    ▲ MOR-T1, An Loch Mór

✖ V1477, Lake Lögurinn    ◼ Lake Getvaltjärnen 9-9.5 cm





**Figure 4: Geochemical characterisation of glass tephra shards from the TUNU2013 78.655 m deposit and comparisons to Icelandic proximal material and analyses of Veiðivötn 1477 CE occurrences. (a) (i) Inset of total alkali versus silica plot showing the tholeiitic basalt composition of the TUNU2013 78.655 m deposit. Chemical classification and nomenclature after Le Maitre et al. (1989). Division line between alkaline and sub-alkaline material from MacDonald and Katsura (1964). (ii and iii) Geochemical characterisation of the TUNU2013 78.655 m deposit compared to geochemical envelopes for several Icelandic tholeiitic volcanic systems. Geochemical envelopes derived from unnormalised whole rock and glass shard analyses from deposits proximal to the Icelandic volcanic systems, adapted from Óladóttir et al. (2020) and references within. Unnormalised TUNU2013 78.655 m data utilised to permit a direct comparison with the published geochemical envelopes. (b) (i-iii) Comparisons of individual analyses from the geochemical characterisation of TUNU2013 78.655 m to characterisations of several Icelandic and non-Icelandic occurrences of the Veiðivötn 1477 CE eruption. Geochemical data for V1477 occurrences from Svartárvatn (Larsen et al., 2002), Helluvaðstjörn (Lawson et al., 2007), Skaftártunga (Streeter and Dugmore, 2014), Hvítárvatn (Larsen et al., 2011), Lake Lögurinn (Gudmundsdóttir et al., 2016), Gripdeild (Bergþórsdóttir, 2014), a compilation of six sites around the Vatnajökull ice cap (Óladóttir et al., 2011), MD99-2275 (Larsen et al., 2002; Gudmundsdóttir et al., 2011), An Loch Mór (Chambers et al., 2004) and Lake Getvaltjärnen (Davies et al., 2007). All data provided in supplementary information and plotted on a normalised anhydrous basis.**



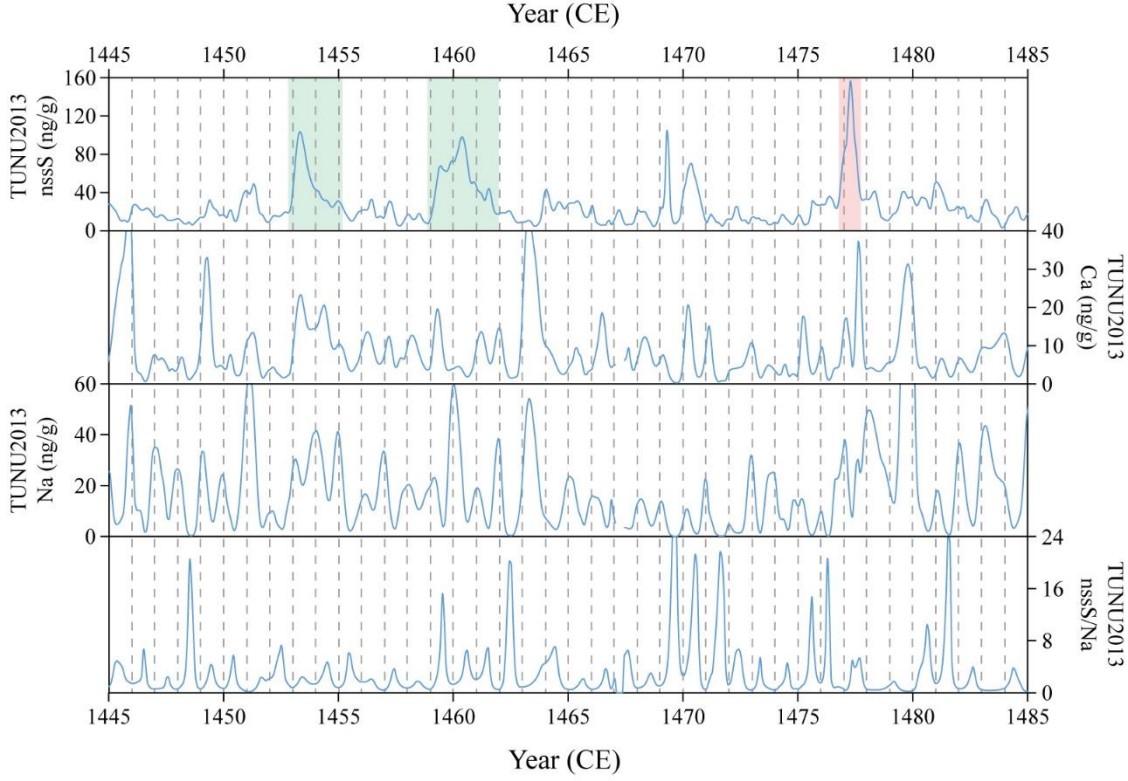

**Figure 5: High-resolution nssS and annual layer dating of the TUNU2013 ice core between 1445–1485 CE using a multiparameter approach. Concentration records are shown for nssS, Ca, Na and nssS/Na with grey dashed lines indicating the assigned years. Red shaded bar indicates the location of the V1477 tephra layer in the TUNU2013 record. Green shaded bars indicate the two 1450s CE sulphate peaks in TUNU2013.**



**Figure 6: (a) Northern hemispheric (40–90°N) summer (JJA) temperature reconstruction depicting the cooling induced by selected**
1040 **15th to 17th century volcanic eruptions. The red line depicts 30-yr smoothed temperatures whereas the blue line indicates annual temperature anomalies. The blue shaded area gives the 95% confidence interval. (b) Cooling induced by the most prominent Icelandic eruptions of the last 1500 years in 939 CE (Eldgjá), 1477 CE (Veiðivötn) and 1783 CE (Laki) in the NVOLC v2 (Guillet et al., 2020), Sch2015 (Schneider et al., 2015) and NTREND2015 (Wilson et al., 2016) as well as in the Scandinavian cluster of NVOLC v2; (c) Spatial patterns of Northern Hemisphere temperatures the year following the three major Icelandic eruptions of**
1045 **the last 1500 years.**