# Peer review of "Cryptotephra from the Icelandic Veiðivötn 1477 CE eruption in a Greenland ice core: confirming the dating of 1450s CE volcanic events and assessing the eruption's climatic impact"

_Climate of the Past, 2020_

## Referee Comment (RC1) · Karen Fontijn (Referee) · 25 Sep 2020

General Comments: Very well written manuscript, with high-quality figures and data presentation. The amount of new data provided is in some ways rather light to deserve a standalone publication (essentially only one tephra layer in one ice core with new data), but perhaps this is common in this field.

Please not that I do not have any expertise in dendrochronology to comment on the methods of Section 3.3 and results of Section 4.3, so hopefully other reviewers will.

[Figure]

Specific Comments (mostly grouped per section):

Abstract: a bit confusing for the novice reader not familiar with the cores in the region. The geographic setting probably needs to be introduced a bit more clearly early in the abstract.

L59-60: with respect to the three papers cited, but "while most of the above-mentioned mismatches have been resolved" seems to be a very bold statement? Please comment some more

Section 2:

- I'm not an expert on Icelandic volcanism as such, but somehow, I have the impression there are a lot of "biggest historical eruptions" and also "one of the most active volcanic systems" in Iceland. It won't really change anything, but some more context could be useful to back up these kinds of statements.

- Can you please elaborate some more on the geochemical fingerprinting of this tephra – if it is basaltic, is it then really that easy to distinguish from other basaltic tephra? What kind of data have the previous chemistry-based correlations mainly been based on (also only major elements on glass, or also other things like trace elements, analyses on crystals)?

- In first approximation tephra will be deposited sooner than the sulphate aerosols (L273), but of course this also depends on the longevity of the eruption. You only comment on this at the very end, in Section 5.5. How detailed are the historical archives? Typically, a fissure eruption may be a long-lived event (except perhaps apart from an intense opening phase that would send particles into the stratosphere). Can you comment on that some more, already in Section 2?

Chemical analysis (Section 3.2):

I am a bit surprised the analyses were performed on an SEM rather than EMP instrument? For major elements modern EDS detectors can indeed be tuned to provide

quantitative data of sufficient quality, but surely not for all elements analysed? You refer to another paper for the more detailed methods, but please elaborate at least a little bit on which elements were analysed using the EDS and which ones using the WDS detector(s), and also on the analytical conditions used (beam size, current, voltage), and how they were adjusted to deal with such fine-grained particles.

Please comment on the appearance of the glass shards, other than their size and brown colour, especially considering the slight mismatch for some elements compared to the previously known 1477CE tephra. Do they contain any microlites, or are they entirely glassy? Any signs of post-depositional alteration?

Samples: Fig 3c suggests the other intervals ii-iii-iv were also sampled for tephra; and these are indeed commented on in Section 4.1.2. It is a bit of a missed opportunity that these were not analysed. Without analysis, it is a bit speculative to discard their correlation to a different event, and simply say they may be remobilised. As commented on earlier: if this were a fissure eruption, could it not be the case that the event was long-lived, and had multiple highly explosive phases? Or is the historical evidence really conclusive that it was not? What other evidence would there be for reworking?

Chemical variability:

L465: if they did indicate a more primitive composition, other elements should also systematically vary consistently with general fractionation trends. Is that really the case?

The two possible explanations given to explain possible chemical variations (Section 5.4), do not seem entirely solid to me (or would need to be justified better): 1. If the two lobes have been identified, how does the chemical composition of the proximal deposits vary, if at all? In other words, do the lobes show any variability? If not, why would the distal deposits in Greenland? 2. If the existing plume of an ongoing eruption were to bifurcate, why/how would it experience chemical fractionation? 3. Alternatively: What is the grain size of the other distal deposits that have previously been analysed?

Is it comparable to the particle size in the studied core, or coarser? In principle one would not really tend to expect variations in glass composition with grain size to be induced by magma fragmentation, but given that there may be at least some local heterogeneity in the melt: is there a possibility that a very fine fraction of particles that happens to be more Mg-rich was preferentially distributed towards Greenland, e.g. due to variability in density?

Technical Comments L136: specify Changbaishan (N Korea) or rephrase – now it reads as if it is in Greenland.

L146: "Volcanic Explosivity Index" – cite Newhall & Self 1982

L210: technically it is not the oxides being analyses, but the elemental concentrations, which are then converted to oxide concentrations.

L307: something wrong in this sentence - remove "were analysed"

L435: for individual shards (bubble walls), vesicularity should not really play a role anymore, so I would suggest removing that.

---

## Referee Comment (RC2) · Anonymous Referee #2 · 20 Oct 2020

General: The paper is well written and finally shines some light on the complicated 1400's volcanic record in Greenland. The figures are very helpful and are well done. Only a few very small things missing. The geochemistry needs some more explanation. Mainly rationale for the analysis type and why the disparity in MgO. Maybe find geochemical data from proximal sources with more geochemical variability. I am not an expert in dendrochronology and supplied general comments but cannot speak to the modeling. With some minor changes, this paper would be a great addition to the Northern Hemisphere volcanic and climate records.

[Figure]

The abstract is long and covers 3 different thoughts that are not tied together well. Overview, characterization of tephra, and then further implications. Could be shorter. IDK why you chose 2500 yrs in the abstract when you only go back to 939 C.E. in your figures. I would remove the text about the coldest summers. The 1477 eruptions did not greatly affect summer temperatures and the text spends too much on things that were found in other studies.

Specific Comments: Line 66- The start of this paragraph does not flow well with the previous paragraph. Could this paragraph be more incorporated into the above paragraph? This paper deals with 2 unknown sulfate spikes (1453 and 1458) and the tephra/sulfate pair you are analyzing. . It would be nice if you introduced them here, individually, rather than lump them as the 1450s. 1453 has the same magnitude but shorter duration than 1458 in TUNU2013 but seems to be working together to cool the 1450's. Line 76- You provide specific locations for climate reconstruction using dendrochronology but then say "northern boreal forest" for Briffa et al., 1998. This could use a little more context. Line 105- I would try to keep things in chronologic order. Talk about 1452 first and then talk about 1458. There is a switch halfway through the sentence that makes it difficult to read. Line 119- add "glass" in front of shards. Want to be clean we are dealing with glass compositions and not mineral compositions. Line 154- Does "historical period" have a specific time frame or is it just the last 1000 years? Line 191- I would define what monthly resolution means here. It shows up later and implies that you know which month the eruption occurred. Monthly at this depth means ~1 cm resolution which you define later in results. Line 205- microscope slide instead of microprobe. A microprobe was not used. Why did you decide to use EDS and WDS instead of a microprobe? You should put analytical conditions in the footnotes of S1a. Beam current, Acc. Voltage, counting times, beam size, etc. It is hard to tell what was analyzed with each detector or the rationale. Could you elaborate? Outside of secondary standards how could someone reproduce this data? Line 272- I would cite Koffman et al., 2013 and Koffman et al., 2017 as they both deal with particle peak to sulfate peak differences in ice. Line 310- I would re-order this paragraph. The most

important metric in this paper is the geochemical correlation of the glass shards. The sulfate offset is empirical and there is no known measurement for calculating that difference into a distance. Line 315- It would be nice to have the geochemistry of these other eruptions made available or discussed more. Line 330- I would move this up to Line 225 and get it out early. I was excited to see geochemistry for these shards. How come they were not analyzed? Could be an interesting story to have MSH in the core. Line 352- What happened to the second coldest summer? Was it not volcanically forced? Line 466- Did you look at more proximal records that show the volcanic succession? It would be nice to see compositions closer to the source. Maybe the more primitive compositions would be there. Maybe comparing the two-lobe directories would be good instead of plotting all the data in Figure 4. The big difference in MgO needs some more explanation. Fig. 1. – I know Sigl et al., 2013 says monthly resolution but it may be easier to use sub-annual as it is hard to see the small variations anyway when looking at 50yrs of data. Fig. 3- Secondary Y-axis says the data is the same but the lines look different. 1477 C.E. particle peak in a) ∼0.10 and in b) ∼0.38. Is the top x-axis correct? Seems like a big just in accumulation change from 78.7-78.6 (∼0.5 yrs) to 78.6-78.5 (∼1 yr.). I only notice the black shard in c). Is that you are referring too or is it all of the smaller clear shards. The dark shard really draws the focus. Might want to add more to this caption. Fig. 5- What is with the sulfate peak at 1469? It has a similar magnitude as 1459 and 1453. Fig. 6- Laki in panel b) is missing the x line denoting where 0 on the temperature anomaly is located. Not all of the blue dots are labeled in panel a). Also not referenced in the caption.

---

## Author Response (AR1)

**Response to Reviewer and Editor Comments for "Cryptotephra from the Icelandic Veiðivötn 1477 CE eruption in a Greenland ice core: confirming the dating of 1450s CE volcanic events and assessing the eruption's climatic impact" by Peter M. Abbott et al.**

**Response to Reviewer 1 Comments**

**General Comments**:

**Reviewer 1:** Very well written manuscript, with high-quality figures and data presentation.

**Response:** We thank the reviewer for these positive comments and the constructive review.

**Reviewer 1:** The amount of new data provided is in some ways rather light to deserve a standalone publication (essentially only one tephra layer in one ice core with new data), but perhaps this is common in this field.

**Response:** We acknowledge that in comparison to other tephrochronological studies, certainly those aiming to build a framework for a set of archives or region, the reporting of a single new occurrence of a tephra horizon could be viewed as light. However, the study and reporting of a single tephra layer (often in only one ice-core) is not uncommon in this field especially when the implications go beyond simply reporting the tephra. For example, when the single tephra layer can aid the dating eruptions or provide insights into past tephra dispersal, or when the discovery is combined with other data such as new geochemical analyses of other occurrences to aid proximal-distal correlations, ice-core glaciochemical signatures of volcanism and reconstructions of the climatic impact of the eruptions. Recent examples of studies of this nature include Jensen et al. (2014), Sun et al. (2014), Dunbar et al. (2017), Cook et al. (2018), Hartman et al., (2019), Narcisi et al. (2019) and McConnell et al. (2020). As our publication goes beyond just reporting this single discovery and utilises the tephra layer to contribute significantly to settling a notable debate regarding the timing and climatic impact of the 1450s CE volcanic eruptions and we also explore the climatic impact of the V1477 eruption we feel it warrants a standalone publication.

**Reviewer 1:** Please not that I do not have any expertise in dendrochronology to comment on the methods of Section 3.3 and results of Section 4.3, so hopefully other reviewers will.

**Specific Comments (mostly grouped per section):**

**Reviewer 1:** Abstract: a bit confusing for the novice reader not familiar with the cores in the region. The geographic setting probably needs to be introduced a bit more clearly early in the abstract.

**Response:** We thank the reviewer for drawing our attention to confusion in the abstract; we have rectified our oversight not to highlight that TUNU2013 is an ice core from Greenland. In addition, reference to the NS1-2011 chronology has been removed from the abstract as readers might not be familiar with this notation.

**Reviewer 1:** L59-60: with respect to the three papers cited, but "while most of the above-mentioned mismatches have been resolved" seems to be a very bold statement? Please comment some more

**Response:** To clarify this statement we have expanded the comment to "While most of the above-mentioned mismatches have been resolved *using new ice-core chronologies and hemispheric-wide tree-ring based climate reconstructions*". We hope this clarifies the statement.

Section 2:

**Reviewer 1:** - I'm not an expert on Icelandic volcanism as such, but somehow, I have the impression there are a lot of "biggest historical eruptions" and also "one of the most active volcanic systems" in Iceland. It won't really change anything, but some more context could be useful to back up these kinds of statements.

**Response:** We have added context that the statement regarding Veidivötn being "one of the most active volcanic systems" is based on the number of eruptions during this period. We don't feel further context is required as this is commonly accepted, and we just wish to provide some context for the V1477 eruption. If further context for this assertion is required, I would direct the reader towards the literature regarding Icelandic volcanism referred to in the manuscript. The same is also true regarding the description of V1477 as one of the most explosive historical eruptions. Where necessary we have added the caveat that it is "one of the biggest/most explosive" not the most explosive as there are many metrics than can be used to judge the size of eruptions, but feel that is sufficient context for the eruption in this manuscript.

**Reviewer 1:** - Can you please elaborate some more on the geochemical fingerprinting of this tephra – if it is basaltic, is it then really that easy to distinguish from other basaltic tephra? What kind of data have the previous chemistry-based correlations mainly been based on (also only major elements on glass, or also other things like trace elements, analyses on crystals)?

**Response:** We have added a comment to Section 2 to clarify that the correlation of V1477 tephra between proximal, medial and distal sites is underpinned by major element analysis of glass shards, but also stratigraphic and geochronological evidence. We also direct the reader to a later section (Section 5.3) where the issue of geochemical similarities between the basaltic products of the Veidivötn system is already highlighted alongside the importance of having chronological evidence to back up any correlations between Veidivötn deposits. We have also highlighted that the correlation between the Veidivötn proximal deposits and tephra layer "a" is based on geochemical evidence and field mapping.

**Reviewer 1:** - In first approximation tephra will be deposited sooner than the sulphate aerosols (L273), but of course this also depends on the longevity of the eruption. You only comment on this at the very end, in Section 5.5. How detailed are the historical archives? Typically, a fissure eruption may be a long-lived event (except perhaps apart from an intense opening phase that would send particles into the stratosphere). Can you comment on that some more, already in Section 2?

**Response:** The record of sulphates and tephra within the ice cores can relate to the longevity of eruptions, but as the reviewer also points out, the explosivity of an eruption is key for tephra particles to be injected high enough into the atmosphere to be transported to Greenland. Based on our review of the literature we have not found any evidence, geological or historical, that the V1477 event was long lasting or had multiple explosive phases. The explosive nature of the event is often highlighted and Larsen et al. (2014) highlight that the tephra fall is correlated to the early stages of the eruption, a point we have now added to Section 2, so we think it is fair to assume the tephra deposition also relates to that phase.

**Reviewer 1:** Chemical analysis (Section 3.2): I am a bit surprised the analyses were performed on an SEM rather than EMP instrument? For major elements modern EDS detectors can indeed be tuned to provide quantitative data of sufficient quality, but surely not for all elements analysed? You refer to another paper for the more detailed methods, but please elaborate at least a little bit on which elements were analysed using the EDS and which ones using the WDS detector(s), and also on the analytical conditions used (beam size, current, voltage), and how they were adjusted to deal with such fine-grained particles.

**Response:** The JEOL FEG-SEM 6500F used for the analysis is a system that combines electron microprobe (field emission gun) and scanning electron microscope capabilities through the addition of a Wavelength Dispersive Spectrometer (WDS) to the SEM. The system allows major element data to

be generated using both WDS and EDS. As shown in the INTAV inter-lab comparison (Kuehn et al. 2011), several labs use integrated WD and ED analysis performed on an SEM for glass geochemical analysis, and Coulter et al. (2010) demonstrated the precision and reliability of the analyses produced by the FEGSEM 6500F and three electron microprobes. Furthermore, our secondary and internal standard glass data illustrate the precision of our analysis. For clarity, we have added details on the operational parameters (including which elements were analysed by WDS or EDS) to the supplementary information and we argue that the secondary glass standard speak to the precision of our results, irrespective of the instrument used for analysis.

**Reviewer 1:** Please comment on the appearance of the glass shards, other than their size and brown colour, especially considering the slight mismatch for some elements compared to the previously known 1477CE tephra. Do they contain any microlites, or are they entirely glassy? Any signs of post-depositional alteration?

**Response:** Most shards are plate-like or occasionally cuspate, and shards containing microlites were present but rare. All shards were sharp edged, none showing any indication of physical alteration. These details have been added to the revised manuscript.

**Reviewer 1:** Samples: Fig 3c suggests the other intervals ii-iii-iv were also sampled for tephra; and these are indeed commented on in Section 4.1.2. It is a bit of a missed opportunity that these were not analysed. Without analysis, it is a bit speculative to discard their correlation to a different event, and simply say they may be remobilised. As commented on earlier: if this were a fissure eruption, could it not be the case that the event was long-lived, and had multiple highly explosive phases? Or is the historical evidence really conclusive that it was not? What other evidence would there be for reworking?

**Response:** The shards from these additional samples have not yet been analysed because of their small size, low concentration and low probability of yielding useful data due to the presence of microlites and flat morphology. We feel that it is appropriate to consider their correlation to the specific event of the 1479 CE tephra of Fiacco et al. (1993) unlikely based on the shards not having physical characteristics (i.e. colourless glass) indicative of the rhyolitic composition characterising that deposit. However, we acknowledge the reviewer's concern that we have limited evidence to attribute it to remobilisation so have adapted our assessment of that potential from "most likely" to "possible" and in the manuscript highlight that further work may help solve this unanswered question. We feel that such work is beyond the scope and focus of this manuscript and would not have an impact on the major conclusions of this work. Based on our review of the literature we have not found any evidence, geological or historical, that the V1477 event was long lasting or had multiple eruptive phases over a period of 2 years. Therefore, we do not feel that the shards identified in the 1479 CE annual layer derived from a later eruptive phase related to V1477.

**Reviewer 1:** Chemical variability:

L465: if they did indicate a more primitive composition, other elements should also systematically vary consistently with general fractionation trends. Is that really the case?

**Response:** We did not clearly observe fractional trends for other element pairs that would support this proposition and have highlighted this in the manuscript. Nevertheless, we feel it is worth highlighting the potential to fully explore all possible explanations for the MgO difference.

**Reviewer 1:** The two possible explanations given to explain possible chemical variations (Section 5.4), do not seem entirely solid to me (or would need to be justified better):

1. If the two lobes have been identified, how does the chemical composition of the proximal deposits vary, if at all? In other words, do the lobes show any variability? If not, why would the distal deposits in Greenland?

**Response:** As we outline in the manuscript geochemical differences have not been identified thus far along different dispersal axes; however, this is primarily based on proximal-medial deposits and there are insufficient data from far distal sites to assess if there is geochemical variation at those sites. In Section 5.4 we aimed to highlight that we cannot provide a definitive conclusion for the geochemical differences and to discuss issues that could be considered in future studies. This was important as the secondary standards did not suggest there was analytical uncertainty. We have now also included analyses of a Veidivötn tephra made at the same time as the TUNU2013 analyses as a QUB internal standard. At the time of the analysis the specific eruption was uncertain, but during revision of the manuscript it has been confirmed as a sample of V1477. These analyses do not display the offset in MgO, providing further evidence that the MgO offset for the TUNU2013 analyses was not due to analytical uncertainty.

**Reviewer 1:** 2. If the existing plume of an ongoing eruption were to bifurcate, why/how would it experience chemical fractionation?

**Response:** We have amended the text to refer specifically to changes in wind direction during the eruption.

**Reviewer 1:** 3. Alternatively: What is the grain size of the other distal deposits that have previously been analysed? Is it comparable to the particle size in the studied core, or coarser? In principle one would not really tend to expect variations in glass composition with grain size to be induced by magma fragmentation, but given that there may be at least some local heterogeneity in the melt: is there a possibility that a very fine fraction of particles that happens to be more Mg-rich was preferentially distributed towards Greenland, e.g. due to variability in density?

**Response:** As grain-size data and more specifically the grain-size of the individual shards analysed for their major element composition has not been reported in the other studies it is not possible to explore this issue, but, we have added a statement to acknowledge that should be considered in the future.

**Technical Comments**

**Reviewer 1:** L136: specify Changbaishan (N Korea) or rephrase – now it reads as if it is in Greenland.

**Response:** Done.

**Reviewer 1:** L146: "Volcanic Explosivity Index" – cite Newhall & Self 1982

**Response:** Done.

**Reviewer 1:** L210: technically it is not the oxides being analyses, but the elemental concentrations, which are then converted to oxide concentrations.

**Response:** Text changed to clarify this point.

**Reviewer 1:** L307: something wrong in this sentence - remove "were analysed"

**Response:** Done.

**Reviewer 1:** L435: for individual shards (bubble walls), vesicularity should not really play a role anymore, so I would suggest removing that.

**Response:** Done.

**References**

Cook, E., Portnyagin, M., Ponomareva, V., Bazanova, L., Svensson, A. and Garbe-Schönberg, D.: First identification of cryptotephra from the Kamchatka Peninsula in a Greenland ice core: Implications of a widespread marker deposit that links Greenland to the Pacific northwest. Quaternary Sci. Rev., 181, 200-206, doi: 10.1016/j.quascirev.2017.11.036, 2018.

Coulter, S.E., Pilcher, J.R., Hall, V.A., Plunkett, G. and Davies, S.M.: Testing the reliability of the JEOL FEGSEM 6500F electron microprobe for quantitative major element analysis of glass shards from rhyolitic tephra. Boreas, 39, 163-169, doi: 10.1111/j.1502-3885.2009.00113.x, 2010.

Dunbar, N.W., Iverson, N.A., Van Eaton, A.R., Sigl, M., Alloway, B.V., Kurbatov, A.V., Mastin, L.G., McConnell, J.R. and Wilson, C.J.N.: New Zealand supereruption provides time marker for the Last Glacial Maximum in Antarctica. Sci. Rep., 7, 12238, doi: 10.1038/s41598-017-11758-0, 2017.

Fiacco, R.J., Palais, J.M., Germani, M.S., Zielinski, G.A. and Mayewski, P.A.: Characteristics and Possible Source of a 1479 A.D. Volcanic Ash Layer in a Greenland Ice Core. Quaternary Res., 39(3), 267-273, doi: 10.1006/qres.1993.1033, 1993.

Hartman, L.H., Kurbatov, A.V., Winski, D.A., Cruz-Uribe, A.M., Davies, S.M., Dunbar, N.W., Iverson, N.A., Aydin, M., Fegyveresi, J.M., Ferris, D.G., Fudge, T.J., Osterberg, E.C., Hargreaves, G.M. and Yates, M.G: Volcanic glass properties from 1459 C.E. volcanic event in South Pole ice core dismiss Kuwae caldera as a potential source. Sci. Rep., 9, 14437, doi: 10.1038/s41598-019-50939-x, 2019.

Jensen, B.J.L., Pyne-O'Donnell, S., Plunkett, G., Froese, D.G., Hughes, P.D.M., Sigl, M., McConnell, J.R., Amesbury, M.J., Blackwell, P.G., van den Bogaard, C., Buck, C.E., Charman, D.J., Clague, J.J., Hall, V.A., Koch, J., Mackay, H., Mallon, G., McColl, L. and Pilcher, J.R.: Transatlantic distribution of the Alaskan White River Ash. Geology, 42(10), 875-878, doi: 10.1130/G35945.1, 2014.

Kuehn, S.C., Froese, D.G. and Shane, P.A.R.: The INTAV intercomparison of electron-beam microanalysis of glass by tephrochronology laboratories: Results and recommendations. Quatern. Int., 246, 19-47, doi: 10.1016/j.quaint.2011.08.022, 2011.

Larsen, G., Eiríksson, J. and Gudmundsdóttir, E.R.: Last millennium dispersal of air-fall tephra and ocean-rafted pumice towards the north Icelandic shelf and the Nordic seas, in: Marine Tephrochronology, edited by: Austin, W.E.N., Abbott, P.M., Davies, S.M., Pearce, N.J.G. and Wastegård, S., Geological Society, London, 113-211, doi: 10.1144/SP398.4, 2014.

McConnell, J. R., Sigl, M., Plunkett, G., Burke, A., Kim, W.M., Raible, C. C., Wilson, A. I., Manning, J. G., Ludlow, F. M., Chellman, N. J., Innes, H. M., Yang, Z., Larsen, J. F., Schaefer, J. R., Kipfstuhl, S., Mojtabavi, S., Wilhelms, F., Opel, T., Meyer, H., and Steffensen, J. P.: Extreme climate after massive eruption of Alaska's Okmok volcano in 43 BCE and effects on the late Roman Republic and Ptolemaic Kingdom, P. Natl. Acad. Sci. USA, 202002722, doi: 10.1073/pnas.2002722117, 2020.

Narcisi, B., Petit, J.R., Delmonte, B., Batanova, V. and Savarino, J.: Multiple sources for tephra from AD 1259 volcanic signal on Antarctic ice cores. Quaternary Sci. Rev., 210, 164-174, doi: 10.1016/j.quascirev.2019.03.005, 2019.

Sun, C., Plunkett, G., Liu, J., Zhao, H., Sigl, M., McConnell, J.R., Pilcher, J.R., Vinther, B., Steffensen, J.P. and Hall, V.: Ash from Changbaishan Millennium eruption recorded in Greenland ice: Implications for determining the eruption's timing and impact. Geophys. Res. Lett., 41(2), 694-701, doi: 10.1002/2013GL058642, 2014.

**Response to Reviewer 2 Comments**

**Reviewer 2:** The paper is well written and finally shines some light on the complicated 1400's volcanic record in Greenland. The figures are very helpful and are well done.

**Response:** We thank the reviewer for these positive comments and the constructive review.

**Reviewer 2:** Only a few very small things missing. The geochemistry needs some more explanation. Mainly rationale for the analysis type and why the disparity in MgO. Maybe find geochemical data from proximal sources with more geochemical variability.

**Response:** We acknowledge that the geochemistry does need some further explanation and hope that our responses to more specific comments from the reviewer address their concerns. The disparity in the MgO was an issue with exploring the correlation between the TUNU2013 tephra and proximal deposits that we fully acknowledged and tried to address. As such, we explored and proposed many possible explanations for the offset and outlined those in the manuscript, but at the present time cannot identify a definitive explanation. As we outline later in the paper this was supported by compiling a comprehensive dataset of V1477 analyses. Further work on characterising proximal deposits may show this geochemical variation. We have now also included analyses of a Veidivötn tephra made at the same time as the TUNU2013 analyses as a QUB internal standard. At the time of the analysis the specific eruption was uncertain, but during revision of the manuscript it has been confirmed as a sample of V1477. These analyses do not display the offset in MgO, providing further evidence that the MgO offset for the TUNU2013 analyses was not due to analytical uncertainty.

**Reviewer 2:** I am not an expert in dendrochronology and supplied general comments but cannot speak to the modeling. With some minor changes, this paper would be a great addition to the Northern Hemisphere volcanic and climate records.

**Response:** We thank the reviewer for this positive comment.

**Reviewer 2:** The abstract is long and covers 3 different thoughts that are not tied together well. Overview, characterization of tephra, and then further implications. Could be shorter. IDK why you chose 2500 yrs in the abstract when you only go back to 939 C.E. in your figures. I would remove the text about the coldest summers. The 1477 eruptions did not greatly affect summer temperatures and the text spends too much on things that were found in other studies.

**Response:** We have tried to improve the abstract to tie the different aspects of this study together. 2,500 years is not the specific time interval for this study, but a general comment regarding the time period over which studies of this nature – annual comparisons between tree-ring and ice-core records – can be conducted. This is due to the availability of well-dated records from both archives, sufficient replication and spatial coverage of tree-ring records and historical evidence for volcanic eruptions during this period. As suggested, we have removed the text regarding the summer temperatures from the abstract.

**Specific Comments:**

**Reviewer 2:** Line 66- The start of this paragraph does not flow well with the previous paragraph. Could this paragraph be more incorporated into the above paragraph? This paper deals with 2 unknown sulfate spikes (1453 and 1458) and the tephra/sulfate pair you are analyzing. . It would be nice if you introduced them here, individually, rather than lump them as the 1450s. 1453 has the same magnitude but shorter duration than 1458 in TUNU2013 but seems to be working together to cool the 1450's.

**Response:** We have tried to improve the clarity of this section. However, as we are trying to describe the evolution of the debate regarding this eruption, we don't introduce the idea of two eruptions until

later. Initially low-resolution ice-core glaciochemical data pointed towards a single event that was attributed to the climatic cooling and the formation of the Kuwae caldera. Subsequently, high-resolution ice-core measurements could resolve annual variations in sulphate, providing evidence for and dating the two eruptions, hence why this issue is introduced later.

**Reviewer 2:** Line 76- You provide specific locations for climate reconstruction using dendrochronology but then say "northern boreal forest" for Briffa et al., 1998. This could use a little more context.

**Response:** We have made it clearer that Briffa et al. (1998) identified the climatic cooling at various sites in their network and that this was "circum-northern hemispheric" in nature.

**Reviewer 2:** Line 105- I would try to keep things in chronologic order. Talk about 1452 first and then talk about 1458. There is a switch halfway through the sentence that makes it difficult to read.

**Response:** Done.

**Reviewer 2:** Line 119- add "glass" in front of shards. Want to be clean we are dealing with glass compositions and not mineral compositions.

**Response:** Done.

**Reviewer 2:** Line 154- Does "historical period" have a specific time frame or is it just the last 1000 years?

**Response:** We have added more context to highlight that the historical period here refers to the time since the settlement of Iceland.

**Reviewer 2:** Line 191- I would define what monthly resolution means here. It shows up later and implies that you know which month the eruption occurred. Monthly at this depth means ~1 cm resolution which you define later in results.

**Response:** We have added a statement to the text to define how we regard the data as monthly, or quasi-monthly, resolution and how it could be best described as sub-annual (see later clarification). Over our period of interest, the measurement resolution is about 1 cm, which for a 12 cm weq/y ice core relates to a quasi-monthly resolution, assuming that snowfall is constant throughout the year. The annual layer boundaries are defined using the nssS/Na ratio (see manuscript Figure 5) based on their opposing seasonality with a winter Na peak and S minimum (Maselli et al., 2017). Estimating the true month of deposition is difficult as errors arise from assigning the annual layer boundary and the seasonal distribution of snowfall. We estimate these to be ± 1 month and ± 2 months respectively and thus estimate the total error on absolute month assignment during this period to be ± 2.2 months.

As such we don't try to define the exact months of deposition for the sulphates and particles, but feel that based on the data we can fairly confidently state that they and thus the eruption occurred early in the year, i.e. winter, and based on the chronology and tephra identification can say it was 1477 CE. We have clarified our statement that the delay between the particle peak deposition and the sulphates was 2 months to "approximately 2 months" due to the uncertainties described above.

**Reviewer 2:** Line 205- microscope slide instead of microprobe. A microprobe was not used. Why did you decide to use EDS and WDS instead of a microprobe? You should put analytical conditions in the footnotes of S1a. Beam current, Acc. Voltage, counting times, beam size, etc. It is hard to tell what was analyzed with each detector or the rationale. Could you elaborate? Outside of secondary standards how could someone reproduce this data?

**Response:** Analyses were conducted in the instrument used at Queen's University Belfast for tephra glass analysis, which has been specifically enabled for precision analysis of fine ash particles, as demonstrated by Coulter et al. (2010). This has been clarified in the manuscript and analytical conditions have been included in the SI. We have added the dimensions of the glass slides used for mounting for clarification, but retain "microprobe slides" as this is a common way to refer to this size of glass size to differentiate them from other sized slides more commonly used for microscope work in tephra studies and shard quantification.

**Reviewer 2:** Line 272- I would cite Koffman et al., 2013 and Koffman et al., 2017 as they both deal with particle peak to sulfate peak differences in ice.

**Response:** Done.

**Reviewer 2:** Line 310- I would re-order this paragraph. The most important metric in this paper is the geochemical correlation of the glass shards. The sulfate offset is empirical and there is no known measurement for calculating that difference into a distance.

**Response:** Done.

**Reviewer 2:** Line 315- It would be nice to have the geochemistry of these other eruptions made available or discussed more.

**Response:** The geochemistry of these other eruptions and their similarities to V1477 are discussed later in the manuscript in Section 5.3. We do not feel it is necessary to present the previously published data further in this manuscript as the high chronological precision of the ice-core chronology implies that the TUNU 78.655 m tephra layer does not have chronological similarities to any of the other Veidivötn eruptions.

**Reviewer 2:** Line 330- I would move this up to Line 225 and get it out early. I was excited to see geochemistry for these shards. How come they were not analyzed? Could be an interesting story to have MSH in the core.

**Response:** We prefer not to move the text. The shards identified in these samples have not yet been analysed because of their small size, low concentration and low probability of yielding useful data due to the presence of microlites and flat morphology. As outlined in the manuscript the physical characteristics of the 1479 CE shards and their similarities to the 1477 CE shards strongly indicate they are of a mafic composition and thus highly unlikely to correlate to the rhyolitic material previously identified by Fiacco et al. (1993) or typically reported from Mount St Helens. Further work is needed for this event, but this is beyond the scope and focus of this manuscript.

**Reviewer 2:** Line 352- What happened to the second coldest summer? Was it not volcanically forced?

**Response:** The second coldest summer in 1699 CE (-1.49°C) is not listed as it was not volcanically forced and has been attributed to natural climate variability during a known cold period, the Maunder Minimum. In addition, there is no sulphate signal in the Greenland records consistent with a large volcanic eruption prior to this cold year.

**Reviewer 2:** Line 466- Did you look at more proximal records that show the volcanic succession? It would be nice to see compositions closer to the source. Maybe the more primitive compositions would be there. Maybe comparing the two-lobe directories would be good instead of plotting all the data in Figure 4. The big difference in MgO needs some more explanation.

**Response:** We conducted a thorough search of all V1477 deposit characterisations reported in the literature and did not identify more primitive compositions comparable to the TUNU2013 78.655 m shards. As we could not identify chemical differences between the lobes and strong similarities

between all V1477 deposits (see similarity coefficient comparisons in Table S5) we don't think it is necessary to split the data in that way. Within Section 5.4 we have explored many potential factors for the difference in MgO to try to find an explanation and feel we cannot provide a definitive explanation at this time.

**Reviewer 2:** Fig. 1. – I know Sigl et al., 2013 says monthly resolution but it may be easier to use sub-annual as it is hard to see the small variations anyway when looking at 50yrs of data.

**Response:** Done.

**Reviewer 2:** Fig. 3- Secondary Y-axis says the data is the same but the lines look different. 1477 C.E. particle peak in a) ~0.10 and in b) ~0.38. Is the top x-axis correct? Seems like a big just in accumulation change from 78.7-78.6 (~0.5 yrs) to 78.6-78.5 (~1 yr.). I only notice the black shard in c). Is that you are referring too or is it all of the smaller clear shards. The dark shard really draws the focus. Might want to add more to this caption.

**Response:** The difference in the peak heights is due to the resolution of the data, with panel (a) showing annual resolution data and panel (b) showing sub-annual data, which mutes the peak heights in the annual data. The same is true for the sulphate data on these plots.

The top axis is correct. The change in accumulation rate at this point is due to the 1477 CE sulphate peak being used as an age marker and fix point in the chronology.

Further description of the glass shard (only one shown) has been included in the figure caption for clarification. Smaller clear shards are not present; the reviewer is perhaps referring to the texture created when the glass slide is frosted to allow the resin to bond to the surface.

**Reviewer 2:** Fig. 5- What is with the sulfate peak at 1469? It has a similar magnitude as 1459 and 1453.

**Response:** The 1469 CE sulphate peak in TUNU2013 is a volcanic sulphate deposition signal (with co-registered increases in acidity and liquid conductivity, not shown) with high concentrations but very short time duration of deposition (<4 nominal months) typical for tropospheric emissions close or upwind of Greenland and with a short atmospheric aerosol lifetime. It is not regarded as a significant event as it is not detected in other Greenland ice cores NEEM-2011-S1 or NGRIP (see manuscript Figure 1) unlike the signals in 1453, 1458/59 and 1477 CE and was not investigated here as there is not an associated microparticle peak. A consistent signal is observed around 1470 CE in several Greenland ice cores (manuscript Figure 1); however, it is not investigated in this study as it was not accompanied by a co-registered microparticle peak (see manuscript Figure 3b).

**Reviewer 2:** Fig. 6- Laki in panel b) is missing the x line denoting where 0 on the temperature anomaly is located. Not all of the blue dots are labeled in panel a). Also not referenced in the caption

**Response:** Done.

**References**

Briffa, K.R., Jones, P.D., Schweingruber, F.H. and Osborn, T.J.: Influence of volcanic eruptions on Northern Hemisphere summer temperature over the past 600 years. Nature, 393, 450-455, doi: 10.1038/30943, 1998.

Coulter, S.E., Pilcher, J.R., Hall, V.A., Plunkett, G. and Davies, S.M.: Testing the reliability of the JEOL FEGSEM 6500F electron microprobe for quantitative major element analysis of glass shards from rhyolitic tephra. Boreas, 39, 163-169, doi: 10.1111/j.1502-3885.2009.00113.x, 2010.

Fiacco, R.J., Palais, J.M., Germani, M.S., Zielinski, G.A. and Mayewski, P.A.: Characteristics and Possible Source of a 1479 A.D. Volcanic Ash Layer in a Greenland Ice Core. Quaternary Res., 39(3), 267-273, doi: 10.1006/qres.1993.1033, 1993.

Koffman, B.G., Dowd, E.G., Osterberg, E.C., Ferris, D.G., Hartman, L.H., Wheatley, S.D., Kurbatov, A.V., Wong, G.J., Markle, B.R., Dunbar, N.W., Kreutz, K.J. and Yates, M.: Rapid transport of ash and sulfate from the 2011 Puyehue-Cordón Caulle (Chile) eruption to West Antarctica. J. Geophys. Res. Atmos., 122, 8908-8920, doi: 10.1002/2017JD026893, 2017.

Koffman, B.G., Kreutz, K.J., Kurbatov, A.V. and Dunbar, N.W.: Impact of known local and tropical volcanic eruptions of the past millennium on the WAIS Divide microparticle record. Geophys. Res. Lett., 40, 4712-4716, doi: 10.1002/grl.50822, 2013.

Maselli, O. J., Chellman, N. J., Grieman, M., Layman, L., McConnell, J. R., Pasteris, D., Rhodes, R. H., Saltzman, E. S. and Sigl, M.: Sea ice and pollution-modulated changes in Greenland ice core methanesulfonate and bromine. Clim. Past, 13, 39-59, doi: 10.5194/cp-13-39-2017, 2017.

**Response to Editor Comments**

We thank the editor for also looking over the submission and providing additional comments that have helped to improve the revised manuscript.

**Editor Comment L96-97:** Here, the 1452/53 and 1458/59 events are linked to NH and SH low latitude origins, respectively. But where does this level of precision come from? Usually, ice core sulfate is used to attribute to one of 3 regions, tropics, NH extratropics or SH extratropics. Toohey et al. (2016) attempted to provide more precise estimate of latitude for the 536 eruption based on model results and identified eruptions, (in supplemental information), but there is a fair amount of variability in both which makes this difficult. In any case, the latitudinal attributions suggested here need evidence: some discussion of the relationships between Antarctic and Greenland sulfate flux would help.

**Response:** Our suggestion that the 1453 event is likely located in the Northern hemisphere low latitudes and the 1458 event likely in the Southern hemisphere low-latitudes is based on three lines of evidence; (1) Simultaneous deposition of sulphate (within relative dating errors) on both polar ice sheets (Sigl et al., 2013; Plummer et al., 2012); (2) isotopic (i.e. $^{33}$S) anomaly in volcanic sulphate indicates stratospheric aerosol formation (Cole-Dai et al., 2013; Gautier et al., 2019) -- both suggest a stratospheric injection outside the high-latitudes; (3) the strong and opposing asymmetry in the deposition of sulphate towards the Greenland and Antarctica ice-sheet, respectively, form the basis for our attribution to the respective hemisphere. While other factors than latitude may influence the distribution of sulphate between both hemispheres, aerosol modelling suggests that a likely latitudinal position of eruption sources can be approximated based on the asymmetry of sulphate deposition (e.g. Toohey et al., 2016; 2019; Marshall et al., 2019). The strong asymmetry of sulphate towards the SH in 1458 (which is rare in the past 2 ka) was used as a key argument in some of the early studies to suggest Kuwae located at 17°S as a potential source volcano (e.g. Gao et al., 2006). To address this comment, we removed the text about our own hemispheric attribution here, but added the arguments outlined above to the Discussion section 5.1 Confirming the timing of mid-15th century volcanic eruptions.

**Editor Comment L107:** by "SH" here, do you mean SH tropical? This is confusing, since, as stated above, most readers would assume a binning into the usual 3 categories, in which case "SH" would be interpreted as SH extratropical.

**Response:** The difficulty in a precise language in describing the hypothesis formulated by Esper et al. (2017) relates to the fact that these colleagues did not mention the earlier sulphate signal in Antarctica in 1453 or the fact that there were two sulphate spikes with the same temporal spacing also in Greenland. They also remain vague with respect to which ice-core chronology they consider to be offset, that of Greenland or that of Antarctica. We made a few changes to the manuscript to clarify this issue.

**Editor Comment L341:** There is inconsistent reference to the 2nd event of the 1450s: sometimes referred to as "1458" and sometimes "1458/59". If the same thing is meant in all cases, it should be made consistent. And if the latter term is used, it should be explained what it means—is the two year range an uncertainty in the eruption year, or does the deposition occur over both years, or something else? (The same thing happens a few times for 1452/53).

**Response:** There are inherent age uncertainties in dating past eruptions using proxy records related to 1) cumulative age uncertainties in the annual layer interpretation of ice cores, 2) the brevity of the growth phase of tree-rings during the Northern Hemisphere summer and 3) the time lag between sulfur emissions at a volcanic source and subsequent deposition of sulphuric aerosols on polar ice-sheets which can range between a few weeks up to a year depending on location and other eruption source parameters. To provide a consistent terminology we labelled unknown volcanic eruptions after

the year in which increase of volcanic sulphate was first detectable in high-resolution ice-cores from Greenland and Antarctica (e.g. 1453 CE and 1458 CE events in Sigl et al., 2013; Plummer et al., 2012). The actual eruption date is either in the same year or the previous year, due to the potential time delay of volcanic sulphate deposition on the ice-core sites (e.g. Marshall et al., 2019). To address this comment we have added sentences addressing the points made above to Sections 1.1 and 1.2 and readjusted the labels throughout the manuscript and on Figs. 1 and 3.

**Editor Comment L354:** please specify whether the 30-year running mean is centered on the year of the calculated anomaly or is the 30 years before the year.

**Response:** The 30-year running mean is centred on the year of the calculated anomaly and this has been described earlier in the manuscript in Section 3.3.1 with the 30-year window around 1477 CE described as an example. To emphasis this and avoid confusion we have reiterated that it is centred around the year of the anomaly in the sentence that was L354 in the original submission.

**Editor Comment L356:** Please double check these numbers, especially the anomaly for SCH2015, in Fig 6b the anomaly for SCH2015 seems to be much closer to zero than -0.35degC (perhaps it's -0.035degC?).

We thank the editor for highlighting this error and have changed to anomaly for Sch 2015 to the correct value of -0.04 °C.

**Editor Comment L385:** Again, by "Northern Hemisphere" do you mean north of the equator but in the tropics?

**Response:** We have expanded the arguments for our attributions (as outlined above regarding the Line 96-97 comment) by adding a paragraph to this section, 5.1, of the manuscript.

**Editor Comment L421:** Miller supports the intensification of the LIA in the 15th Century, but not the inception: they suggest the mid-13th Century for the inception. Schurer's definition of the LIA to start in 1450 seems to be directly following the IPCC AR5 report, which provides little objective reasoning for the date. The best I know of is Owens et al. (2017).

**Response:** We decided to leave the term inception, because any boundary is arbitrarily in a constantly changing climate. Miller et al. (2012) provides proxy information from a comparably small geographical region in the Arctic, whereas Owens et al. (2017) and Büntgen et al. (2020) collated data over larger geographical areas. Both of these two studies define the 1440s or 1450s as the inception of the LIA. We added the references to Owens et al. (2017) and Büntgen et al. (2020), following the suggestion of the editor.

**Editor Comment L487:** It's relatively well known that the VEI is not a great predictor of climate impact, see the discussion in Robock 2000. The VEI is primarily a measure of the volume of tephra emitted, while the climate impact is most closely related to the amount of sulfur injected into the stratosphere. These two quantities can be loosely correlated, but aren't going to scale directly for all eruptions. In various places throughout the manuscript, V1477 is described as very "explosive". To a reader like me, this implies a rapid eruption rate, and vigorous eruption plume and likely stratospheric injection. But this might not be the case. It would be helpful if the term "explosive" is defined in the manuscript, especially in regards to the V1477 eruption. Is there evidence of a high eruption plume? Does the fact that it was a fissure eruption (of substantial length) reduce the probability of a strong eruption plume? Or does the description of the eruption as "most explosive" (line 155) strictly refer to the VEI value? (If the last, I would suggest being specific and referring to the eruption as having the largest VEI).

**Response:** This is exactly the point we are making with this comparison: VEI is a poor predictor of the climate impact potential of past eruptions and the amount and height of sulfuric gas emissions

remains the most important parameter. VEI is nevertheless, the most widespread used information accessible for many eruptions in the geologic record as it combines information about the intensity of the eruption with the magnitude. Therefore, we consider it important to demonstrate that eruptions with a smaller VEI (such as Laki and Eldgja) can have a larger impact on climate then those with a high VEI. We added information about eruption conditions, intensity and plume height into Section 2.

**Editor Comment L495:** Downwelling over the polar cap during winter is rather slow compared to horizontal mixing. Toohey et al. (2019) found a small difference in radiative forcing between winter and summer high latitude, lower stratospheric eruption simulations. Season of eruption may be part of the story, but there's not much evidence to suggest the season is a major factor in explaining the difference between V1477 and eruptions like Laki. To me, most likely is that there is significant uncertainty in the stratospheric sulfur injection for the different eruptions based on the ice cores.

**Response:** We appreciate the feedback on stratospheric dynamics and previous work on winter vs. summer radiative impacts of volcanic eruptions. We have removed the last sentence.

**References**

Büntgen, U., Arseneault, D., Boucher, É., Churakova, O. V., Gennaretti, F., Crivellaro, A., Hughes, M. K., Kirdyanov, A. V., Klippel, L., Krusic, P. J., Linderholm, H. W., Ljungqvist, F. C., Ludescher, J., McCormick, M., Myglan, V. S., Nicolussi, K., Piermattei, A., Oppenheimer, C., Reinig, F., Sigl, M., Vaganov, E. A., and Esper, J.: Prominent role of volcanism in Common Era climate variability and human history, Dendrochronologia, 64, 125757, 2020.

Cole-Dai, J., Ferris, D.G., Lanciki, A.L., Savarino, J., Thiemens, M.H. and McConnell, J.R.: Two likely stratospheric volcanic eruptions in the 1450s C.E. found in a bipolar, subannually dated 800 year ice core record. J. Geophys. Res., 118, 7459-7466, doi: 10.1002/jgrd.50587, 2013.

Esper, J., Büntgen, U., Hartl-Meier, C., Oppenheimer, C. and Schneider, L.: Northern Hemisphere temperature anomalies during the 1450s period of ambiguous volcanic forcing. Bull. Volcanol., 79, 41, doi: 10.1007/s00445-017-1125-9, 2017.

Gao, C., Robock, A., Self, S., Witter, J.B., Steffensen, J.P., Clausen, H.B., Siggaard-Andersen, M.-L., Johnsen, S.J., Mayewski, P.A. and Ammann, C.: The 1452 or 1453 A.D. Kuwae eruption signal derived from multiple ice core records: Greatest volcanic sulfate event of the past 700 years. J. Geophys. Res., 111(D12), D12107, doi: 10.1029/2005JD006710, 2006.

Gautier, E., Savarino, J., Hoek, J., Erbland, J., Caillon, N., Hattori, S., Yoshida, N., Albalat, E., Albarede, F. and Farquhar, J.: 2600-years of stratospheric volcanism through sulfate isotopes. Nat. Commun., 10, 466, doi: 10.1038/s41467-019-08357-0, 2019.

Global Volcanism Program, 2013. Volcanoes of the World, v. 4.9.1 (17 Sep 2020). Venzke, E (ed.). Smithsonian Institution. Downloaded 04 Dec 2020

Larsen, G., Eiriksson, J., and Gudmundsdottir, E. R.: Last millennium dispersal of air-fall tephra and ocean-rafted pumice towards the north Icelandic shelf and the Nordic seas. In: Marine Tephrochronology, Austin, W. E. N., Abbott, P. M., Davies, S. M., Pearce, N. J. G., and Wastegard, S. (Eds.), Geological Society Special Publication, 2014.

Larsen, G.: Explosive Volcanism in Iceland: Three Examples of Hydromagmatic Basaltic Eruptions on long Volcanic Fissures within the past 1200 Years; Geophysical Research Abstracts, Vol. 7, 10158, 2005

Marshall, L., Johnson, J.S., Mann, G.W., Lee, L., Dhomse, S.S., Regayre, L., Yoshioka, M., Carslaw, K.S., and Schmidt, A.: Exploring How Eruptive Source Parameters Affect Volcanic Radiative Forcing Using Statistical Emulation. J. Geophys. Res., 124(2), 964-985, doi: 10.1029/2018JD028675, 2019.

Miller, G. H., Geirsdóttir, A., Zhong, Y. F., Larsen, D. J., Otto-Bliesner, B. L., Holland, M. M., Bailey, D. A., Refsnider, K. A., Lehman, S. J., Southon, J. R., Anderson, C., Björnsson, H. and Thordarson, T.: Abrupt onset of the Little Ice Age triggered by volcanism and sustained by sea-ice/ocean feedbacks. Geophys. Res. Lett., 39(2), L02708, doi: 10.1029/2011GL050168, 2012.

Owens, M. J., Lockwood, M., Hawkins, E., Usoskin, I., Jones, G. S., Barnard, L., Schurer, A. and Fasullo, J.: The Maunder minimum and the Little Ice Age: an update from recent reconstructions and climate simulations, J. Sp. Weather Sp. Clim., 7, A33, doi:10.1051/swsc/2017034, 2017.

Plummer, C.T., Curran, M.A.J., van Ommen, T.D., Rasmussen, S.O., Moy, A.D., Vance, T.R., Clausen, H.B., Vinther, B.M. and Mayewski, P.A.: An independently dated 2000-yr volcanic record from Law Dome, East Antarctica, including a new perspective on the dating of the 1450s CE eruption of Kuwae, Vanuatu. Clim. Past., 8, 1929-1940, doi: 10.5194/cp-8-1929-2012, 2012.

Sigl, M., McConnell, J.R., Layman, L., Maselli, O., McGwire, K., Pasteris, D., Dahl-Jensen, D., Steffensen, J.P., Vinther, B., Edwards, R., Mulvaney, R. and Kipfstuhl, S.: A new bipolar ice core record of volcanism from WAIS Divide and NEEM and implications for climate forcing of the last 2000 years. J. Geophys. Res., 118(3), 1151-1169, doi: 10.1029/2012JD018603, 2013.

Toohey, M., Krüger, K., Schmidt, H., Timmreck, C., Sigl, M., Stoffel, M. and Wilson, R.: Disproportionately strong climate forcing from extratropical explosive volcanic eruptions, Nat. Geosci., 12(2), 100–107, doi:10.1038/s41561-018-0286-2, 2019.

Toohey, M., Krüger, K., Sigl, M., Stordal, F. and Svensen, H.: Climatic and societal impacts of a volcanic double event at the dawn of the Middle Ages, Clim. Change, 136(3–4), 401–412, doi:10.1007/s10584-016-1648-7, 2016.